

**Atmo-metabolomics: a new measurement approach for investigating aerosol composition**
**and ecosystem functioning.**
Albert Rivas-Ubach[1,2],  Yina Liu[1], Jordi Sardans[2,3], Malak M. Tfaily[1], Young-Mo Kim[4], Eric
Bourrianne[5],  Ljiljana Paša-Tolić [1], Josep Peñuelas[2,3], Alex Guenther[6].
1. Environmental Molecular Sciences Laboratory, Pacific Northwest National Laboratory, Richland,
99354, WA, USA.
2. CREAF, Cerdanyola del Vallès, 08913 Catalonia, Spain
3. CSIC, Global Ecology Unit CREAF-CEAB-CSIC-UAB, Cerdanyola del Vallès, 08913 Catalonia, Spain
4. Biological Sciences Division, Pacific Northwest National Laboratory, Richland, 99354, WA, USA.
5. Faculté des Sicences et d'Ingénierie, Université de Toulouse III Paul Sabatier, Toulouse, 31400, France.
6. Department of Earth System Science, University of California, Irvine, CA, USA
**Author of correspondence:**
Albert Rivas-Ubach
Environmental Molecular Sciences Laboratory,
Pacific Northwest National Laboratory,
Richland, WA, USA, 99354
Tel: 971 319 5962
e-mail: albert.rivas.ubach@gmail.com




**Abstract.**
Aerosols directly and indirectly play crucial roles in the processes controlling the composition
of the atmosphere and the functioning of ecosystems. Gaining a deeper understanding of the
chemical composition of aerosols is one of the major challenges for atmospheric and climate
scientists and is beginning to be recognized as important for ecological research. Better
comprehension of aerosol chemistry can potentially provide valuable information on
atmospheric processes such as oxidation of organics and the production of cloud condensation
nuclei as well as provide an approximation of the general status of an ecosystem through the
measurement of certain stress biomarkers. In this study, we describe an efficient aerosol
sampling method, the metabolite extraction procedures for the chemical characterization of
aerosols, namely, the atmo-metabolome. We used mass spectrometry (MS) coupled to liquid
chromatography (LC-MS), gas chromatography (GC-MS) and Fourier transform ion cyclotron
resonance (FT-ICR-MS) for a deep characterization of the atmo-metabolome. The atmo-
metabolomes from two distinct seasons, spring and summer, were compared to test the
sensitivity and demonstrate the information that can be provided from each analytical
platform. Our results showed that our sampling and extraction methods are suitable for
aerosol chemical characterization with any of the analytical platforms used in this study. The
three datasets obtained from these individual platforms showed significant differences of the
overall atmo-metabolome between spring and summer. LC-MS and GC-MS analyses identified
several metabolites that can be attributed to pollen and other plant-related aerosols. Spring
samples exhibit higher concentrations of metabolites linked to higher plant activity while
summer samples had higher concentrations of metabolites that may reflect certain oxidative
stresses. FT-ICR-MS analysis showed clear differences in the elemental composition of aerosols
between spring and summer. Summer aerosols were generally higher in molecular weight and
with higher O/C ratios, indicating higher oxidation levels and condensation of compounds
relative to spring. Our method represents an advanced approach for characterizing the
composition of aerosols that will benefit scientists attempting to understand complex
atmospheric processes and the ecosystem status across a whole ecoregion.










## 1. Introduction

### 1.1 Atmo-metabolomics.

Metabolomics aims to study the metabolome of entire organisms or specific cells or tissues. A metabolome consists of the thousands of small (< 1,000 Da) molecular weight compounds (metabolites) present in an organism at a given time (Fiehn, 2002). Such molecules include the substrates and products of cellular primary metabolism such as sugars, amino acids, and nucleotides as well as the plant and fungi secondary metabolism compounds such as terpenoids. They are all involved in a great variety of complex physiological processes to maintain the organisms' homeostasis, growth and responses to biotic and non-biotic stressors (Peñuelas and Sardans, 2009). The metabolomes can thus be considered as the chemical phenotype of organisms (Fiehn, 2002). Metabolomic techniques have been widely applied in biomedicine (Claudino et al., 2007; Walsh et al., 2006), human nutrition (Gibney et al., 2005; Wishart, 2008), plant physiology (Hirai et al., 2004; Kaplan et al., 2004), and more recently in ecology (ecometabolomics) (Bundy et al., 2008; Rivas-Ubach et al., 2012; Sardans et al., 2011) to understand how metabolomes change under certain circumstances or stressors. Additionally, metabolomics has recently been demonstrated to be a valuable tool for understanding the metabolome plasticity of wild organisms under different environmental situations (Gargallo-Garriga et al., 2014; Rivas-Ubach et al., 2016a, 2016c; Sardans et al., 2011, 2014).

Aerosol is a gaseous suspension of solids and/or liquids (Canagaratna et al., 2007) and can be derived from both biogenic and anthropogenic sources. Aerosols can directly and indirectly influence the atmospheric processes (Carlton et al., 2010; Després et al., 2012; Ramanathan et al., 2001), leading to potentially strong feedbacks on the hydrological cycle and climate (Andreae and Rosenfeld, 2008; Spracklen et al., 2011). Primary biological aerosol particles (PBAP) are directly released from organisms and include cells such as pollen, spores, or whole microorganisms as well as fragments from plants and animal debris (Després et al., 2012). Plants also produce large amounts of volatile organic compounds (VOCs) which are emitted into the atmosphere and together with anthropogenic sources, such as the combustion of fossil fuels, are oxidized and then condense forming secondary organic aerosols (SOA) (Després et al., 2012; Fuzzi et al., 2006; Pandis et al., 1992) (Figure 1). In addition to the effects on atmospheric processes and climate, the deposition of aerosols can directly interact with aquatic and terrestrial ecosystems (Baker et al., 2003; Gu et al., 2002; Mahowald et al., 2005; Seco et al., 2007) by being absorbed by plants (Wedding et al., 1975) and by serving as an important carbon and nutrient source for the microbial communities coexisting in plant



leaves; the phyllosphere (Vorholt, 2012). Aerosols are also of interest because of their
importance for human health including lung diseases and allergies (D'Amato et al., 2002;
Després et al., 2012; Pope et al., 1995) (Figure 1).

In this study, we propose for the first time the application of metabolomic techniques

to the study of the molecular composition of aerosols. We refer to this method as atmo-
metabolomics here and onward. To the best of our knowledge, such an approach has not
previously been reported elsewhere. This novel method of combining atmospheric sampling
and metabolomic analyses provides useful information for ecologists, atmospheric scientists
and even other disciplines such as allergology. Ecologists can benefit from this novel approach
for investigating the response of whole ecosystems, and even whole ecoregions, to
environmental changes. Plants have shown large chemical composition shifts when submitted
to environmental stressors (Leiss et al., 2009; Macedo, 2012; Rivas-Ubach et al., 2014, 2016a;
Robertson, 2005; Sardans et al., 2011) and those changes should also be reflected in the
chemistry of aerosols. Furthermore, several biogenic compounds present in the atmosphere,
such as terpenes, are directly related to plant anti-stress mechanisms (Peñuelas and Llusià,
2001) and such compounds can ultimately condense and contribute to the SOA pool. Recent
climate projections predict an enhancement of extreme climatic events such as warming and
drought which will lead to increases in plant stress and BVOC emissions (Peñuelas and Staudt,
2010) and atmo-metabolomics may serve as a powerful tool to assess such stress at ecosystem
and larger scales. The application of atmo-metabolomics in natural ecosystems represents a
new approach that complements existing aerosol analysis techniques byusing the metabolic
composition in aerosols as an indication of ecological status of the whole ecosystem. Atmo-
metabolomics could provide a valuable measurement approach for following the dynamics of
ecological status in response to natural and anthropic environmental changes. It would also
benefit atmospheric scientists that require innovative tools to identify and quantify the
immense diversity of biogenic contributions to the composition of the atmosphere (Guenther,
2013). Aerosol composition plays an important role in air quality and climate change and there
is an urgent need to improve model simulations of their sources and atmospheric impacts
(Hoyle et al., 2009). For example, revealing the aerosol chemical composition is necessary to
understand atmospheric processes such as new particle formation (Andreae and Crutzen, 1997;
Zhang et al., 2004), formation of cloud condensation nuclei (Ayers and Gras, 1991; Jokinen et
al., 2015) and ice nucleation (Baustian et al., 2012). A major challenge in applying
metabolomics analyses to atmospheric aerosol is the confounding effects of atmospheric
processing, including oxidation and deposition, on the original emission profile. However,
there may be an opportunity to improve understanding of atmospheric processing using this



observational approach if the influence of these processes is recognized and can be
characterized. Additionally, there is also a critical need to characterize the diversity of aerosol
composition in order to predict current and future impacts on human health (Pöschl and
Shiraiwa, 2015).

We propose, therefore, to define the metabolome of the air (atmo-metabolome) as

the total set of molecules in the atmosphere of an area for a period of time including the
particle phase composed of PBAP, SOA and, anthropogenic aerosols as well as the gas phase
including BVOCs (Figure 1).  The techniques to characterize the gas phase component of atmo-
metabolomes are well described elsewhere (Smith and Španěl, 2011; Tholl et al., 2006). Our
purpose here is to describe an atmo-metabolomic method for sampling aerosols and
characterize the particle phase of the atmo-metabolomes.

**1.2 Applying metabolomics techniques to characterize aerosol chemical composition.**
There are several methodologies that can be used to characterize the metabolome of a sample.
Nuclear magnetic resonance (NMR) and mass spectrometry (MS) coupled to liquid or gas
chromatographs (LC-MS and GC-MS respectively) are the most common instruments for
metabolomic analyses (Fiehn, 2002; Sardans et al., 2011; Zhang et al., 2012). Although NMR-
based metabolomics has proven to be very reproducible and quantitative, its sensitivity for
detecting compounds is very low compared to MS techniques (Pan and Raftery, 2007). Given
the low concentrations of atmospheric aerosols, our study is focused only on the classic LC-MS
and GC-MS techniques due to their higher sensitivity relative to NMR. In addition to LC-MS and
GC-MS, we also report aerosol data from Fourier transform ion cyclotron resonance mass
spectrometry (FT-ICR-MS) that can very accurately provide important molecular information of
aerosols (Brown et al., 2005). Nonetheless, no single mass spectrometry technique can cover
all metabolite classes (Ding et al., 2007; Zhang et al., 2012), and the combination of platforms
is a common approach in metabolomics to increase the number of metabolites measured in
the metabolomes (Hall, 2006).

LC-MS and GC-MS techniques provide a similar data format (or dimension) in

metabolomic studies; i.e. in both techniques, metabolites are first separated through
chromatography (liquid or gas) resulting in two independent and orthogonal values; mass-to-
charge ratio (m/z) and retention time (RT) relative to each of the ions detected which are used
to further improve the metabolite assignation (Sumner et al., 2007). Moreover,
chromatography improves the chances to uncover metabolites in small concentrations or even
novel metabolites by decreasing mass spectra complexity at a given RT (Farag et al., 2012).
Generally, GC-MS is suitable for detecting compounds from primary metabolism such as


carbohydrates, fatty acids, essential oils, carotenoids and also organic acids (Gullberg et al.,
2004). LC-MS can cover plant secondary metabolites such as flavonoids, alkaloids, phenolic
acids, and saponins together with primary metabolites such as amino acids, carbohydrates and
organic acids (De Vos et al., 2007). The metabolite species identified with each instrument,
however, will depend on the compounds included in the library used for each of the platforms.

Improving the performance of metabolite assignation is one of the main challenges of

MS-based metabolomics, for this reason the mass resolving power of the spectrometers is an
important factor to consider. The modern Orbitrap mass spectrometers achieve resolutions up
to 140,000 (Roberg-Larsen et al., 2015; Weber et al., 2011) which reduces considerably the
error of metabolite matching when using high-resolution metabolite libraries that include the
exact mass and RT information of the compounds (Rivas-Ubach et al., 2016b). FT-ICR-MS
affords the highest mass resolving power (up to 1,000,000) enabling thus the formula
calculation of a wide range of detected ions (Marshall et al., 1998). Although FT-ICR-MS can be
coupled to liquid chromatography, direct infusion ESI (DI) is the most common method to
analyze samples with this technique. DI only provides m/z of the detected ions, but the
ultrahigh resolution of the FT-ICR-MS makes this a powerful research technique  to understand
the global characteristics of any complex organic samples (Kim et al., 2003; Reemtsma, 2009;
Roullier-Gall et al., 2014; Schmitt-Kopplin et al., 2012; Sleighter and Hatcher, 2007; Tfaily et al.,
2015). In addition, the ultrahigh mass resolution ( < 1 ppm mass error after internal calibration)
enables accurate elemental formula assignments to most of the detected ions based on their
exact mass alone (Klein et al., 2006; Kujawinski, 2002). DI-FT-ICR-MS alone is not sufficient for
putative metabolite identification, and further verification should be performed using MS/MS
fragmentation or NMR (Sumner et al., 2007), however, one significant advantage of generating
the molecular formulas by DI-FT-ICR-MS spectra is the possibility of quantifying the number of
molecular species with different essential nutrients such as nitrogen, phosphorus or sulfur.
This is especially interesting to understand how the elemental assignation in aerosols shift in
response to environmental changes; an important issue for ecological stoichiometry studies
(Rivas-Ubach et al., 2012; Sardans et al., 2012; Sterner and Elser, 2002).

The high mass measurement accuracy of FT-ICR-MS instruments allows confident

elemental formula assignments of the detected ions (Kujawinski and Bhen 2006), and thereby
enable chemical characteristic visualization using van Krevelen diagrams (vK) (Kim et al., 2003;
van Krevelen, 1950). vK diagrams were initially proposed to study the evolution of oils and coal
samples (Curiale and Gibling, 1994; Hatcher et al., 1989; van Krevelen, 1950), however,
plotting O/C vs H/C ratios of all of the assigned formulas of the ions in natural organic matter
(NOM) samples can also provide a useful approximation of the compound classes present in



the samples (Kim et al., 2003; Sleighter and Hatcher, 2007). Such a classification has been
widely use in NOM characterization studies (Kim et al., 2003; Roullier-Gall et al., 2014;
Sleighter and Hatcher, 2007; Tfaily et al., 2015). Moreover, vK diagrams can also be very useful
for atmospheric sciences since it provides information on reactions such as methylation,
demethylation, hydrogenation, hydration, condensation, oxidation or reduction of the
detected ions (Kim et al., 2003). Other graphical representations such as C number versus m/z
(CvM) provide crucial information on the oxidation or the structural size of molecular
compounds when comparing two or more systems (Reemtsma, 2009). Thus, FT-ICR-MS is  a
very useful tool to gain a better understanding of the aerosol sources as well as their chemical
transformation in the atmosphere.

**1.3 Initial case study.**
We present the application of the atmo-metabolomics technique by showing results of aerosol
composition from an initial case study that contrasts two distinct seasons: spring and summer.
We designed a simple aerosol sampling method and collected total aerosol particles (without
any size cutoff) in spring and summer of 2015 at the Pacific Northwest National Laboratory
campus (Richland, WA, USA). We used those samples to describe an operational protocol to
extract the metabolites from aerosols to posteriorly obtain the metabolome fingerprints with; i)
LC-MS, ii) GC-MS and iii) DI-FT-ICR-MS. The generated data with each of the instruments was
analyzed following some basic statistical approximations typical for metabolomics and
chemical characterization studies. The aerosol sampling method, the metabolite extraction
procedures and the main metabolomic differences between spring and summer are discussed.
Although we describe specific procedures and analyses, we also emphasize the flexibility of our
method for different or more specific purposes. Additionally, this method can be adapted for
experimental aerosol chambers for laboratory studies.


**2. Experimental details.**

**2.1 Study site.**
Sampling was conducted at the Pacific Northwest National Laboratory (PNNL) campus (46° 34′
N, 119° 28′ W) located in the north side of the city of Richland (Washington, USA). Nearby
landscape is a desert mainly covered by shrubs and steppes with *Ericameria nauseosa*,
*Chrysothamnus viscidiflorus, Purshia tridentate, Grayia spinose, Artemisia tripartita,*
*Sarcobatus vermiculatous*, *Salsola tragus* and, *Tamarix romosissima* as some of the common



species. The PNNL campus is covered by lawn and introduced planted tree species such as
*Platanus sp.* The surrounding metropolitan area has a population of about 250,000 and the
economy and land use is dominated by agriculture and the nearby Hanford nuclear reservation.
The climate is semi-arid desert with a mean annual precipitation ranging between 180 and 220
mm per year. Annual thermic amplitude is large with an average maximum annual
temperature around 32°C, with peaks reaching up to 42-45°C and the average minimum
annual temperature is -2°C with lowest peaks reaching temperatures of -20°C.

**2.2 Aerosol sampling.**
To represent the spring season, we sampled aerosols in 2015 from May 7th to 20th, both
inclusive (14 consecutive days). For the summer season, samples were collected in 2015 from
July 15th to 30th, both inclusive (16 consecutive days). According to weather conditions
reported by the US National Weather Service at the local airport (KPSC),  the May sampling
period had daily average (maximum) temperature ranging from 11 to 21°C (14 to 29°C) and
daily average (maximum) humidity ranging from 49 to 78% (72 to 100%) while the July
sampling period had daily average (maximum) temperature ranging from 19 to 29°C (28 to
40°C) and daily average (maximum) humidity ranging from 35 to 50% (57 to 86%). Total
precipitation of 28.2 mm was reported for the May sampling period and no precipitation was
reported for the July sampling period. For the aerosol collection, we designed a simple and
portable aerosol sampling system that allows the sampling of multiple filters at once (Figure 2).
Aerosol particles were collected on Whatman QM-A 37mm high-purity quartz filters (Whatman
International Ltd, Maidstone, UK), which were precombusted for 5hrs at 450°C to minimize
any impurity (Schmitt-Kopplin et al., 2012). Two filters were simultaneously collected each day.
A precombusted quartz filter was inserted into a filter cassette. Each of the cassettes were
previously slightly modified from the commercial type to optimize them for our purpose;
briefly, 12 extra holes of 1.5 mm of diameter were placed homogenously on the surface where
the filters are placed to ensure a better distribution of the air along the surface of the filter
(Figure 2a) (a small circular grille could be also used for this purpose). Also, we changed the
position of the different sections of the cassette in order to obtain an air camera between the
filter and tube connector to ensure equality in the air suction from each of the extra holes
(Figure 2b), this was achieved by placing the top section of the cassette at the bottom so that
the filters were totally open to the exterior but sustained by a piece of the cassette (Figure 2c).
Filter cassettes were connected to the pump by using PVC flexible tubing of 0.6 cm diameter
(Figure 2d). The pump was working daily during 18 consecutive hours and pumped air at 30 L
per minute through each filter. Filters were replaced manually before 09:00am and the pump





started working automatically at 09:00am and stopped automatically at 03:00am the following
day. Filters were stored at -80°C until metabolite extraction. Filters were sampled on a tower
at 8 meters height.

One of the objectives of this study was to describe an operational protocol to extract the

metabolites from aerosols and posteriorly analyze with the corresponding instruments. The
extraction of metabolites was mainly sonication-based, so an additional aerosol sampling was
performed in late spring to test different sonication times during the extraction of polar and
semi-polar metabolites and analyzed by LC-MS and GC-MS analyses. For that, we sampled 3
filters during two consecutive days at a flow rate of 30L per minute (18 hours of sampling per
day) (hereafter test-filters). We sampled 6 rounds of test-filters (3 filters x 6 rounds = 18 filters).
The pump started sampling at 09:00am and stopped at 03:00am each day. Sampling was
performed from June 5$^{th}$ to the 16$^{th}$ (12 days). Filters were also stored at -80°C until metabolite
extraction.

**2.3 Metabolite extraction for mass spectrometry analysis.**

Three different tube sets were labeled; set A (8mL glass tubes) to perform the

extractions, set B (15 mL polypropylene centrifuge tubes) to keep the extracts and set C (2 mL
glass tubes) to keep the concentrated extract. Each filter was carefully rolled (Figure 3.1) and
introduced into the corresponding tube of set A (Figure 3.2). Five mL of MeOH/H$_2$O (80:20)
was added as an extraction solvent (this volume of extract was enough to cover the 37mm
filters but it may vary depending on the diameter of the set A tubes) (Figure 3.3) and samples
were sonicated for 10 min at 24ºC (Figure 3.4). For each tube of set A, 4 mL of the extract was
transferred to the corresponding 15 mL centrifuge tubes of set B (Figure 3.4.1). These
procedures were repeated on the same filters to perform two extractions but adding 4 mL of
MeOH/H$_2$O (80:20) as fresh extract and the resulting extract was thus combined with the initial
one (Figures 3.5, 3.5.1). All extracts in tubes of set B were then dried with an ultra-high purity
nitrogen evaporator (Figure 3.6) and 1 mL of fresh extraction solvent was posteriorly added to
each tube and vortexed for 30 s to ensure the correct dissolution of the extract (Figure 3.7).
Tubes of set B were thus centrifuged for 5 min at 4,000 x g (Figure 3.8) and supernatants were
transferred into the set C of 2 mL glass tubes (Figure 3.9). Samples were then stored at -80 ºC
until the mass spectrometry analysis (Figure 3.10).

The extracts were analyzed by LC-MS (Orbitrap mass analyser), GC-MS (single

quadrupole mass analyzer) and DI-FT-ICR-MS (12T) (Figure 3.11). For DI and LC-MS analyses;
the extracts from all samples were directly introduced into a labeled HPLC vial set with inserts



(Figure 3.12). We typically add 200 µL of extract in the HPLC but this volume may be varied for
other studies.

GC-MS required a pre-treatment of the samples prior to the instrumental analyses; the

dried extracted metabolites were chemically derivatized to their trimethylsilyl ester forms as
previously described (Kim et al., 2015). For the derivatization, first 500 µL of each extract from
the set of tubes C (Figure 3.10) were placed into a set of glass vials and dried down in a
vacuum evaporator. Once dried, 20 µL of methoxyamine in pyridine (30 mg/mL) was added to
each sample. All vials were vortexed for 30 seconds and incubated at 37°C in a Thermomixer
(Eppendorf AG, Hamburg, Germany) for 90 min with shaking at 1000 rpm to protect carbonyl
groups. After the first incubation, all samples were centrifuged for 15 seconds and 80 µL of N-
methyl-N-(trimethylsilyl)trifluoroacetamide (MSTFA) with 1% trimethylchlorosilane (TMCS)
was added to each vial. Vials were then vortexed for 10 seconds and again incubated for 30
min at 37°C with shaking (1,000 rpm) to derivatize hydroxyl, carboxyl and amine groups. After
the second incubation, vials were centrifuged for 15 seconds and extracts were transferred
into clean labeled glass vials with 200 µL inserts by using Pasteur pipettes. A cap with septum
was then tightened onto each of the vials.
The description of the method used to test different sonication times during metabolite
extraction is detailed in the supporting information (Supplementary Text).

**2.4 LC-MS analysis.**
LC-MS chromatograms were obtained using a Vanquish ultra-high pressure liquid
chromatography (UHPLC) system coupled to an LTQ Orbitrap Velos high-resolution mass
spectrometer equipped with a heated electrospray ionization (HESI) source (Thermo Fisher
Scientific, Waltham, Massachusetts, USA). A reversed-phase C18 Hypersil gold column (150 ×
2.1 mm, 3µ particle size; Thermo Scientific, Waltham, Massachusetts, USA) at 30 ºC was used.
The mobile phases consisted of acetonitrile (A) and water (0.1% acetic acid) (B). Mobile phases
were filtered and degassed for 15 min in an ultrasonic bath prior to use. At a flow rate of 0.3
mL per minute, the elution gradient initiated at 10% A (90% B) and was held for 5 min, then
the gradient linearly changed to 10% B (90% A) for the next 15 min. The initial proportions (10%
A; 90% B) were thus linearly recovered over the next 5 min, and the column was washed and
stabilized for 5 more minutes. The injection volume of the samples was 5 µL. All samples were
analyzed in both positive (+) and negative (-) ionization modes. The Orbitrap mass
spectrometer was operated in FTMS (Fourier Transform Mass Spectrometry) full-scan mode
with a mass range of 50-1000 m/z at 60,000 resolving power. Blank samples were analyzed





during the sequence and a mixture of standards at known concentration were injected every
15 samples to test instrument sensitivity and mass accuracy.

**2.5 GC-MS analyses.**
After derivatization, samples were cooled down to room temperature and posteriorly analyzed
by an Agilent GC 7890A coupled with MSD 5975C mass spectrometer (Agilent Technologies,
Santa Clara, CA). Separations were performed on a HP-5MS column (30 m × 0.25 mm × 0.25
µm; Agilent Technologies). The injection mode was split-less, and the injection port
temperature was held at 250°C. The column oven was initially maintained at 60°C for 1 min
and then ramped to 325°C by 10°C/min, followed by a 10 min hold at 325°C. Blank controls
and mixture of fatty acid methyl esters (FAMEs; C8-C28) were analyzed prior to sample
analysis.


**2.6 DI-FT-ICR-MS analyses.**
Aerosol extracts were analyzed on a 12 Tesla Bruker SolariX Fourier transform ion cyclotron
resonance (FT-ICR) mass spectrometer (Bruker daltonics Inc, Billerica, MA, USA). Samples were
directly infused into the mass spectrometer using a standard Bruker electrospray ionization
(ESI) in negative mode at a flow rate of 3.0 µL/min through an Agilent 1200 series pump
(Agilent Technologies, Santa Clara, CA, USA) . The ESI source was equipped with a fused silica
tube (30 µm i.d.). The ion accumulation time was optimized for all samples (0.1s). All samples
were analyzed at a resolving power of 400,000 ($m/\Delta m_{50\%}$ at $m/z$ 400). Experimental conditions
were as follows: needle voltage, +4.4 kV; Q1 set to 50 $m/z$; and the heated resistively coated
glass capillary operated at 180 °C.

**2.7 Processing of LC-MS chromatograms.**
The LC-MS files were processed by MZmine 2.17 (Pluskal et al., 2010). Chromatograms of both
positive and negative modes were separately baseline corrected, deconvoluted, aligned and
metabolites were autoassigned before the numerical database was exported in CSV format.
The parameters used for the extraction of the data are given in Table. S1.
Metabolite assignation with LC-MS was performed by our metabolite library with more
than 200 typical metabolites usually present in plants and fungi including products from
primary and secondary metabolism. Assignation were performed separately for each
ionization mode (positive and negative) and using the exact mass of metabolites, their most
abundant fragments and RT. For more detailed information regarding the metabolite matching





see Rivas-Ubach et al., (2016b). RT and m/z values of metabolite matching for LC-MS are
shown in Table S2.

**2.8 Processing of GC-MS chromatograms.**
GC-MS data was processed with two different software; MZmine and Metabolite Detector.
MZmine 2.17 (Pluskal et al., 2010) was specifically used to obtain the metabolomic fingerprints
from the additional sampled filters to test the sonication time and be thus more consistent
with the LC-MS data. Parameters to get the numerical datasets with MZmine are shown in
Table S3.

Metabolite Detector 2.5 (Hiller et al., 2009) was used to process the GC-MS raw data

files from the spring and summer. First, "Agilent .D" files were converted to netCDF format
using Agilent Chemstation and posteriorly converted to "bin" files using Metabolite Detector.
Chromatograms were deconvoluted, aligned and the metabolites were autoassigned before
exporting the datasets in CSV format. Briefly, retention indices (RI) of detected metabolites
were calculated based on the analysis of the FAMEs mixture, followed by their
chromatographic alignment across all analyses after deconvolution. Metabolites were initially
identified by matching experimental spectra to PNNL increased version of FiehnLib (Kind et al.,
2009), containing spectra and validated retention indices for over 850 metabolites, with
probability threshold of 0.8. NIST14 GC-MS library was also used to cross-validate
identification of metabolites by matching fragmented spectra. All metabolite identifications
were manually validated to reduce deconvolution errors during automated data-processing
and to eliminate false identifications. Parameters used in Metabolite detector are shown in
table S4. Metabolite matching information in GC-MS is shown in Table S5.

**2.9 Processing of DI-FT-ICR spectra.**
The mass spectrum for each sample was averaged over 144 individual scans and then
internally calibrated using an organic matter homologous series separated by 14 Da ($-CH_2$
groups). The mass measurement accuracy was typically within 1 ppm for singly charged ions
across a broad $m/z$ range (100-1100 $m/z$). DataAnalysis software (BrukerDaltonik version 4.2)
was used to convert raw spectra to a list of m/z values applying FTMS peak picker with signal
to noise (S/N) threshold of 7 and absolute intensity threshold of 100. Chemical formulas ,
containing C, H, O, N, S, and P, were then assigned using an in-house built software following
the Compound Identification Algorithm (CIA), described by Kujawinski and Behn, (2006) and
modified by Minor et al., (2012). Chemical formulas were assigned based on the following
criteria: S/N >7, mass measurement error <1 ppm. All observed ions in the spectra were singly



charged as confirmed by the 1.0034 Da spacing found between isotopic forms of the same
molecule (i.e., between $^{12}C_n$ and $^{12}C_{n-1}$–$^{13}C_1$).

**2.10 Statistical analyses.**
Overall metabolome fingerprints from aerosols of spring and summer were tested by
PERMANOVAs using the Bray Curtis distance for each dataset generated by LC-MS, GC-MS and
DI-FT-ICR-MS and setting the permutations at 10,000 (Table 1). Posteriorly, the same
metabolome fingerprints were also subjected to principal component analysis (PCA), the most
frequently performed ordination analysis for metabolomics studies to show the natural
variability among the samples reduced typically to two single dimensions (van den Berg et al.,
2006; Kim et al., 2010) (Figure 4).

Heat-map plots for the assigned variables with LC-MS and GC-MS were plotted to show

any metabolite shifts between the two seasons (Figure 5). Each assigned variable was also
submitted to t-student tests with season as the categorical factor (Table S6).

The proportion of each compound class was calculated for each sample by dividing the

number of peaks detected in each compound region by the total number of peaks observed.
We further counted the number of formula classes from the FT-ICR-MS dataset (CHO, CHNO,
CHOS, CHNOS, CHNOSP, CHOSP, CHOP, CHNOP, CHNOPS and CHOPS) for each sample. As
performed with the compound classes, the proportion of each compound class was also
calculated for each sample. All calculated proportions (formula and compound) were
transformed using *arcsin(rootsquare)* before submitting them separately to t-tests with season
(spring and summer) as the categorical factor to investigate whether the presented sampling
and extraction method can statistically discern spring from summer in some of those classes of
formulas and compounds (Figure 6).

The PERMANOVAs, PCAs, heat maps and t-tests were performed with R (R Core Team,

2013). The PERMANOVA analysis was conducted with the *adonis* function in the package
"vegan" (Oksanen et al., 2013). The PCAs were performed by the *pca* function of the
"mixOmics" package of R (Dejean et al., 2013). Heat maps were performed by the *heatmap.2*
function of the "gplots" package (Warnes et al., 2016). T-tests were performed with the
function *t.test* in the package "stats" (R Core Team, 2013). All graphs were obtained by R and
graphically treated by Adobe Illustrator CS6.

The value obtained from the deconvoluted peaks in LC-MS and GC-MS are directly related

to the concentration of the corresponding variable even though they do not represent the real
concentration in the sample in terms of mg of metabolite per weight of sample. However, the
use of those values are suitable for metabolomic comparative analyses as previously



demonstrated in other studies (Gargallo-Garriga et al., 2014; Lee and Fiehn, 2013; Leiss et al.,
2013; Mari et al., 2013; Rivas-Ubach et al., 2014, 2016a). In this study, we use the term *relative*
*abundance* when referring to the relative concentration of metabolites.
FT-ICR data is typically not directly quantifiable (Wozniak et al., 2008), however although
not as robust than LC-MS or GC-MS techniques, using the intensity of the detected ions by FT-
ICR is still a good proxy of their relative concentration (Kellerman et al., 2014; Spencer et al.,
2015). We used the measured ion intensity for the specific vK and CvM representations, for
those purposes the measured intensity of each individual ion detected in each of the samples
was divided by the total intensity of the spectra (Kellerman et al., 2014; Spencer et al., 2015).
For vK and CvM plots, we only used the formula assigned features that presented less than
0.3ppm of error although cutoff values up to 0.5ppm showed good results (Osterholz et al.,

2016).

Chromatograms and spectra from LC-MS and FT-ICR-MS, respectively, of samples
corresponding to days 16[th] and 30[th] June showed signs of contamination and were thus not
considered in the corresponding datasets for statistical analyses.


**3. Results.**

PERMANOVAs of all atmo-metabolome fingerprints generated from each analytical instrument
(LC-MS, GC-MS and DI-FT-ICR-MS) showed significant differences between spring and summer
(Pseudo-F = 2.96, *P* < 0.05; Pseudo-F = 4.41, *P* < 0.0001; and Pseudo-F = 6.46, *P* < 0.001;
respectively) (Table 1).
Accordingly with the results of the PERMANOVA, all performed PCAs with the atmo-
metabolome fingerprints obtained by each instrument showed clear separation between
seasons (Figure 4). The principal component 1 (PC1) and PC2 of the PCA performed with LC-MS
data explained 13.0% and 10.3% respectively of the total metabolomic variance among
samples. The PC1 and PC2 of the PCA performed with GC-MS data explained 24.2% and 13.2%
respectively of the total variance. The PC1 and PC2 from DI-FT-ICR-MS PCA explained 28.2 and
12.9% respectively of the total variance of metabolomes among samples. All PCAs performed
with each mass spectrometry technique showed similar values for the axis that separate
mainly spring and summer cases, being the PC1 for LC-MS Orbitrap (13.0%) and PC2 for GC-MS
and DI-FT-ICR-MS techniques (13.2% and 12.9% respectively).
Student t-tests showed statistical significance between spring and summer in several of
the assigned metabolites with LC-MS and GC-MS (Figure 5 and Table S6). For the dataset



generated by LC-MS, we found that spring had significantly higher relative abundance ($P < 0.05$)
of α-ketoglutaric acid, adonitol, sorbitol-Mannitol, malic acid and marginally higher relative
abundance ($P < 0.1$) of proline, d-tocopherol and hexoses (Figure 5a). Summer had higher
relative abundance of isoleucine ($P < 0.05$) and marginally higher relative abundance of
phenylalanine and coumaric acid ($P < 0.1$). The analyses on the dataset generated by GC-MS
showed that spring had significantly higher relative abundances of glucose and galactose ($P <$
$0.05$) and marginally higher concentrations of trehalose ($P < 0.1$). Fumaric acid was found in
marginally higher relative abundance in the summer ($P < 0.1$) (Figure 5b).

The proportions of CHO, CHNOS, CHOSP and, CHOP formula classes changed significantly

between seasons ($P < 0.05$) (Figure 6a,b). Atmo-metabolomes of spring had significantly higher
proportions of CHNOS, CHOSP and CHOP and marginally higher proportions of CHOS and
CHNOSP. Summer atmo-metabolomes showed higher proportions of CHO than spring.

We found several unique features present in spring and summer aerosols. According to the

compound classification based on Kim et al., (2003) and Sleighter and Hatcher, (2007), we
found that summer aerosols presented more variety of protein-like, lipid-like and amino-sugar
compounds. On the other hand, spring aerosols were characterized by condensed
hydrocarbons and lignin-like compounds (Figure 7a). After plotting the relative intensity
difference of spring with respect to summer of all the detected features assigned to a
molecular formula with less than 0.3ppm of error into a vK diagram, we detected 3 main
regions; two of them with higher relative intensities in spring aerosols and one more intense in
summer (Figure 7b). Based on the compound classification of formula-assigned features,
spring had generally higher relative intensities of condensed hydrocarbons, lignin-like
compounds and carbohydrates than summer. Features detected in summer aerosols
presented higher relative intensities in the protein-like and amino-sugar areas (Figure 7b).

We generally measured higher relative intensities in high-mass features in summer

aerosols with respect to spring aerosols which presented higher relative intensities in lower-
mass features (Figure 8a). Moreover, summer had higher relative intensities of features with
higher-mass than spring but with the same number of C (see region between dashed lines in
Figure 8a). Summer also presented more features with higher number of C than spring. T-test
on the O/C values of the formula-assigned features with season as categorical factor showed
how summer had significantly higher relative intensities in features with higher O/C ratios than
spring (Figure 8b, c).

**4. Discussion.**



**4.1 Aerosol sampling in filters and study site.**

Our sampling method allowed the efficient collection of atmospheric particles on filters. Our simple system consisted of a high-flow oil-free pump, tubing, tube connector fittings, quartz filters and filter-cassettes (Figure 2). This system is highly versatile for various purposes, economic and portable allowing sampling in remote areas if sufficient power is available.

It is important to have optimal flow rates for the aerosol collection; excessive flow rates may collapse the filters and low flow rates will not collect enough particles. Filter sampling should be designed to collect as much aerosol as possible for a good metabolomic analyses performance. We used 37mm quartz filters that performed well without collapsing at flow rates of 50 L/min. Quartz filters from other manufacturers and filters made of other materials such as polytetrafluoroethylene (PTFE) may present different resistances, so it is necessary to know the resistance of the filters used. Since one of the aims of this study was to test the sensitivity of the different mass spectrometry instruments for atmo-metabolomic analyses, our samples were collected at only 30 L/min.

For statistical purposes, the number of biological replicates can be increased by connecting multiple filters at the same time in a specific area. Furthermore, sampling can be performed at different heights on a tower or mast by extending tubing. However, the internal tubing friction associated with the extension of the tubing causes a decrease in in the flow rates at the aerosol collection point. We used 6mm internal diameter tubing but larger diameter can be used to decrease the internal friction in order to increase the flow rate. The performance characteristics of the pump is also a key consideration in the development of a feasible experimental design (including pump flow rate, number of replicates, filter material, length and diameter of tubing) and to maximize the flow rate (~40-45 L/min) at the sampling point for optimum metabolomic analyses. Additionally, most pumps operate at fixed flow rates and stopcock valve can be used to adjust the flow if necessary.

The area surrounding the sampling site should be well characterized in order to interpret correctly the atmo-metabolomic results. In our case, as described in the study site section, aerosol collection was performed in a semi-urban area surrounded by landscapes dominated by large and diverse agricultural cropland and a large desert shrubland with low biological activity, so we expected to detect a complex variety of molecules that complicate finding the atmospheric/ecological interpretation of the data. However, the main aim of this study was to test the sensitivity of different mass spectrometry techniques (LC-MS, GC-MS, FT-ICR-MS) to characterize the metabolomes of aerosols in low activity ecosystems and assess their potential for detecting overall significant changes between seasons. Additionally, wind can transport aerosols and biological particles hundreds of kilometers from their origin (Uno et

al., 2009), so different meteorological variables such as wind speed and direction or rainfall are
important factors to consider since the collected aerosols may potentially correspond to
different ecosystemic scales (local, regional).  For that reason, in ecological comparative
studies, sampling in extensive homogeneous areas facilitates the interpretation of the results
since it minimizes the complexity associated with the mixing of aerosols from multiple source
locations. For example, sampling inside the canopy of a forest can decrease the contribution of
aerosols from distant ecosystems in the analyses. Our system also allows the collection of
aerosols from experimental chambers; for this purpose, researchers should ensure that the
filters will contain enough particle mass for metabolomic analyses.

**4.2 Metabolite extraction in organic solvents.**
Organic solvents combined with water are typically used for metabolomics analyses allowing
the extraction of a good range of semi-polar and non-polar metabolites (Kim et al., 2010; Lin et
al., 2006; Rivas-Ubach et al., 2013; t'Kindt et al., 2008). Solvents such as methanol, acetonitrile
or chloroform interact with plastics, especially under sonication, and chromatograms may
show contaminant features when using plastic tubes for metabolite extraction (Figure S1). The
use of silanized glass tubes are highly recommended during the sonication step (Figure 3.4) to
obtain cleaner extracts without artifacts. Combusted glassware for 5 hours at 450ºC or higher
is also recommended to prevent from any organic contaminants. If plastic tubes are finally
used during the extraction, especially during sonication, an initial test to detect any potential
plastic contaminant is recommended.

Polar and semi-polar metabolites experience large changes in wild plants under

environmental changes (Gargallo-Garriga et al., 2014; Rivas-Ubach et al., 2012, 2014).
Methanol/water (80:20) solution typically showed large polar and semi-polar metabolite
recovery compared to other organic solvents (t'Kindt et al., 2008) and its use in atmo-
metabolomics is suggested but not exclusive. Two or even three extractions, instead of only
one, can be performed on the same sample to increase the metabolite recovery. We
performed two extractions sequentially as detailed in several studies (Böttcher et al., 2007;
Nikiforova et al., 2005; Rivas-Ubach et al., 2013, 2014) (Figure 3.5). Other metabolomic
protocols suggest performing a single extraction to reduce labor time (t'Kindt et al., 2008),
however it will finally depend on the nature of the sample, the concentration, the solvents
used and the procedures performed. Because of the extremely low metabolite concentration
in aerosol samples; we performed two extractions to ensure higher metabolite recovery.

The filter size is also an important factor to consider for atmo-metabolomic analyses.

On one hand, the lower the ratio of *filter size/pump flow rate* is, the more concentrated the



samples will be, allowing better performance by the analytical instruments. On the other hand,
smaller filters are easier to handle in the laboratory during extractions allowing also higher
extract recovery. Quartz filters absorb high extract proportions that cannot be easily recovered.
Our protocol with 37mm diameter filters had an extract recovery of 89%. Larger filters will
complicate the extraction of metabolites (larger tubes and larger volumes of extract required
and probably more filter handling) and decrease considerably the recovery of extracts due the
large solvent absorption.

**4.3 Mass spectrometry instruments.**
We sampled the aerosols in a heterogeneous ecosystem that is impacted by human activities
that contribute to aerosol emissions over the year and is surrounded by a large desert
landscape with relatively low primary production. Even so, LC-MS, GC-MS and, DI-FT-ICR-MS,
demonstrated enough sensitivity to detect overall significant changes in the chemical
composition of aerosols between seasons (Table1 and Figure 4). However, it is important to
note that each technique is not exclusive but complementary since they provide different
information (Ding et al., 2007; Zhang et al., 2012).

GC analyses present excellent reproducibility with minimal RT shifts of the same

detected ions among different samples; however, GC-MS requires sample derivatization which
increases the labor time in sample preparation. Additionally, the instability of the reagents
used for the derivatization process is substantial, so samples should not be kept for long
periods of time and need to be injected shortly after their preparation. Due to the
derivatization of metabolites, GC-MS provide indirect detection of the metabolites that
complicates the elucidation of novel metabolites by ion fragmentation ($MS^n$) without an
appropriate database. LC techniques often show greater RT shifts among samples but samples
derivatization is not required providing thus a direct detection of the metabolites. Even so, if
using high resolution mass spectrometers such as Orbitrap, then metabolite matching can rely
more on the exact mass rather than on RT reducing thus the metabolite matching error and
allowing a more flexible RT error in metabolite matching with LC-MS (Rivas-Ubach et al.,
2016b).

We could match efficiently thousands of molecular formulas present in aerosols with

the DI-FT-ICR data according to C, H, O, N, P and S elements providing important information
on the elemental composition of aerosols (Figure 6). DI-FT-ICR-MS acquisition time is
significantly shorter than MS coupled to a LC or GC. For typical analysis, data acquisition time
for DI-FT-ICR-MS is commonly 5 to 15 minutes per sample on the platform and method used in
this study, while it can take over 40 minutes per sample for LC or GC analyses.  Even so, it is



important to consider that matching the molecular formulas to specific metabolites without
chromatographic separations is challenging because it is not possible to distinguish between
structural isomers. Furthermore, as mass of ions rises, possible structures associated to the
assigned formulas increase substantially too and thus complicates compound structure
identification without chromatography RT, $MS^n$ fragmentation, and standard verifications
(Sumner et al., 2007).

**4.4 Application of atmo-metabolomics to a case study.**
We detected significant overall differences between spring and summer atmo-metabolomes
(Table 1) and clear separation of cases in the PCA (Figure 4) of the metabolome fingerprints
obtained by LC-MS, GC-MS, and DI-FT-ICR-MS which implies that all the detected fractions of
the samples shifted between seasons.
LC-MS and GC-MS detected spring aerosols with higher relative abundance of
carbohydrates such as hexoses, glucose, galactose, trehalose and several other organic acids
related to the tricarboxylic acid cycle such as ketoglutaric acid, malic acid and citric acid (Figure
5) which are good indicators of growth activity in plants (Rivas-Ubach et al., 2012) and
atmospheric pollination (Roulston and Cane, 2000). Those results are in agreement with the
DI-FT-ICR data showing higher proportions of CHOP and CHNOSP molecular formulas (Figure 6)
and higher relative intensities in carbohydrate related compounds (Figure 7). Phosphorus and
carbohydrates have been typically related to higher activity in plants (Rivas-Ubach et al., 2012;
Sterner and Elser, 2002) although sugars can play other functions such as stress tolerance
(Ingram and Bartels, 1996; Rivas-Ubach et al., 2014, 2016c). LC-MS showed that atmo-
metabolomes in summer had higher relative abundance of coumaric acid and acacetin but also
of phenylalanine and shikimic acid tended to be slightly higher in summer (Figure 5a). Shikimic
acid is the precursor of several secondary metabolites such as flavonoids, tannins and other
phenolic metabolites with strong antioxidant activity through phenylalanine and other routes
(Ghasemzadeh and Ghasemzadeh, 2011; Seigler, 1998; Talapatra and Talapatra, 2015).
Antioxidants protect cell membranes from peroxidation (Kim et al., 2005; Rice-Evans et al.,
1996) and have been typically reported to be in higher concentrations in plants under
oxidation stressors such as drought (Peñuelas et al., 2004). Summer is the driest season in the
sampled area receiving up to 3 times less precipitation than spring, for this reason we expect
higher antioxidant activity in plants facing drought stress (Rivas-Ubach et al., 2014, 2016c).
GC-MS detected several fatty acid compounds in the extracts (Figure 5b). Fatty acids
are present in pollen as up to 20% of their dry weight depending on the plant species (Roulston
and Cane, 2000) and arachidic acid and linoleic acid, among others, are typical fatty acids



found in pollen (Solberg and Remedios, 1980). Even though none of the identified fatty acids
showed statistically significant changes between seasons, their relative abundance clearly
tended to increase in spring (Figure 5b), the most active season for plants.

Our FT-ICR analyses showed differences in aerosol composition between spring and

summer (Figures 6 and 7). Although not statistically significant, the slightly higher proportions
of CHNO features found in summer aerosols are in accordance with the higher relative
intensities and more unique features in protein-like compounds found in summer (Figure 7).
Furthermore, summer had significantly higher proportions of CHO features than in spring
(Figure 7) and, in addition, the summer CHO features occur at higher masses than in spring
(Figure 8a). In a CvM plot, mass increase at the same carbon number is account for by the
presence of heteroatoms (e.g. N,S, and O). We observed how features with the same number
of Carbon tended to be at higher m/z values in summer compared to spring (see area between
dashed lined in Figure 8a). Plotting O/C of the CHO formula-assigned features versus mass
revealed that that summer aerosols had higher relative intensities of oxidized compounds than
spring (Figures 8b and 8c) in accordance with the higher compound masses found in summer
respect to spring for a same C-number (Figure 8a). The observed trend suggests that aerosol
components in summer have higher oxidation rates which could be due to higher levels of
photochemical oxidants associated with warm sunny conditions and increased atmospheric
photo-oxidation of aerosols (Obee and Hay, 1997). Moreover, we also found higher relative
intensities in high-mass aerosol compounds (over 500 Da) in summer (Figure 8a) which may
suggest higher rates of polymerization or aerosol condensation. These observations point to
one of the major challenges in utilizing atmo-metabolomic data which is the confounding
effects of atmospheric processing of the original biogenic emissions.

Global change drivers such as warming have been proven to shift the phenology of

plants (Peñuelas and Filella, 2001) and the emissions of BVOCS (Peñuelas and Staudt, 2010),
that could result in significant seasonal changes in the atmo-metabolome of an ecosystem. The
establishment of long term atmo-metabolomic experiments would help with the detection of
significant phenological shifts of entire ecosystems. Furthermore, the use of atmo-
metabolomic techniques in the atmospheric and ecological sciences could improve the
detection, identification and quantification of any molecular compound related with
environmental stressors (biomarkers) (Wolfender et al., 2009) providing thus crucial
information of the general status of the ecosystems. Conversely, changes in the atmo-
metabolomes may trigger impacts on ecosystems and humans (Figure 1).

**4.5 Conclusions.**



· Our portable and low-cost sampling system demonstrated good performance for collecting
atmospheric aerosol samples.
· Although the sampling was performed in a complex region with an urban area surrounded by
a rural desert landscape with relatively low biological activity, all mass spectrometry
techniques (LC-MS, GC-MS and DI-FT-ICR-MS) were still able to detect significant differences
between the spring and summer aerosol metabolomes though the methanol/water (80:20)
extraction.
· There is no unique analytical technique able to characterize the whole metabolome
fingerprint of aerosols. LC-MS and GC-MS and the use of metabolite libraries allow us to detect
specific molecular compounds in aerosols while DI-FT-ICR-MS allows obtaining quickly a high-
resolution metabolic fingerprint providing the elemental composition of aerosol compounds.
· All three analytical techniques showed spring atmo-metabolomes with higher proportions of
carbohydrates and organic acids which is in accordance with the higher biological growth
activity of plants during that season.

**4.6 Future perspectives.**
· Long term atmo-metabolomic experiments in natural ecosystems would improve
understanding of the seasonal and interannual shifts of the composition of aerosols, directly
linking atmospheric composition with plant physiology, along natural gradients or
environmental changes.
· The application of metabolomics to aerosol samples allows the identification of specific
molecular compounds (biomarkers) directly related with specific stressors impacting entire
ecosystems. A good description of such biomarkers and other relevant metabolites would
allow the creation of aerosol compound libraries which could be applied to understand the
status of ecosystems and provide a relatively simple and quick environmental assessment and
monitoring tool.
· Atmo-metabolomics is a promising tool for the identification and determination of the
diversity of the biogenic contribution to atmospheric composition that potentially plays a
crucial role in climate change and air quality.
· New modern instruments such as GC-MS Orbitrap should be implemented in atmo-
metabolomic studies to enable high performance for both RT and m/z resolution. Advances in
methodologies for metabolomic analyses, such as Ion Mobility Spectrometry coupled to mass
spectrometers (IMS-MS), could potentially improve significantly the number of detected
metabolites in aerosols from the current tens and hundreds to thousands.





**Acknowledgements.**
The authors thank Therese Clauss and Rosalie Chu for their laboratory support. This research
was performed using EMSL, a DOE Office of Science User Facility sponsored by the Office of
Biological and Environmental Research at Pacific Northwest National Laboratory and by the
European Research Council Synergy grant SyG-2013-610028 IMBALANCE-P, the Spanish
Government projects CGL2013-48074-P and the Catalan Government project SGR 2014-274.























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





**Table 1.** PERMANOVAs of the atmo-metabolome fingerprints generated by LC-MS, GC-MS and
FT-ICR instruments for overall metabolome comparison between seasons.

| LC-MS | | Sum of Squares | Mean Square | F | *P* |
|---|---|---|---|---|---|
| Season | 1 | 0.65 | 0.65 | 4.41 | 0.0001 |
| Residuals | 26 | 3.82 | 0.15 | | |
| Total | 27 | 4.47 | | | |
| GC-MS | | Sum of Squares | Mean Square | F | *P* |
| Season | 1 | 0.18 | 0.18 | 6.46 | 0.0003 |
| Residuals | 28 | 0.77 | 0.03 | | |
| Total | 29 | 0.94 | | | |
| FT-ICR | Df | Sum of Squares | Mean Square | F | *P* |
| Season | 1 | 0.1145 | 0.11 | 2.96 | 0.0285 |
| Residuals | 26 | 1.01 | 0.04 | | |
| Total | 27 | 1.12 | | | |
























**Figure 1.** Schematic diagram showing the emissions of aerosols and posterior deposition on
ecosystems.

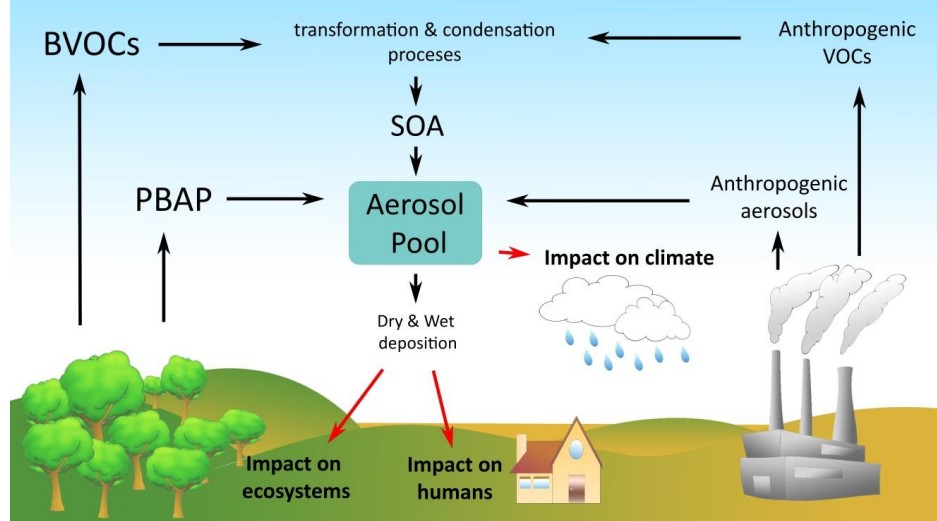





**Figure 2.** Schematic representation of the system used for sampling aerosols.


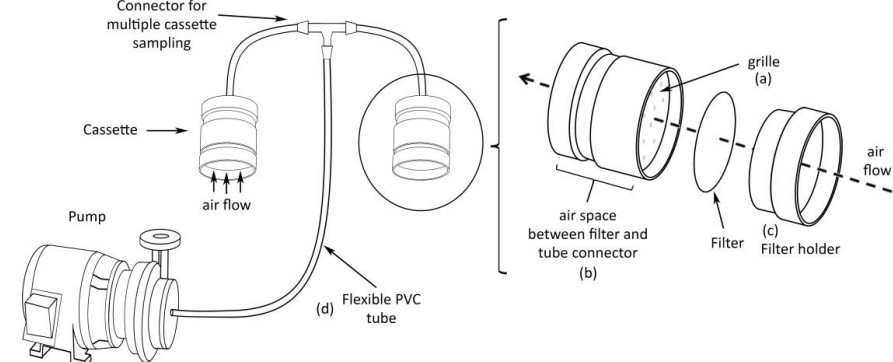


























**Figure 3.** Experimental procedures performed on quartz filters to obtain the semi-polar extracts from aerosols and posteriorly analyze with mass spectrometry techniques.




**Figure 4.** Case plots of the PC1 versus the PC2 of the PCAs conducted from metabolomic
fingerprints of aerosols obtained by LC-MS Orbitrap (LC-MS), GC single quadrupole (GC-MS)
and direct infusion DI-FT-ICR-MS. Each day of sampling correspond to a different point for each
of the graphs. Aerosol metabolomes of spring days are represented by blue triangles and
summer days are represented by red cycles circles.

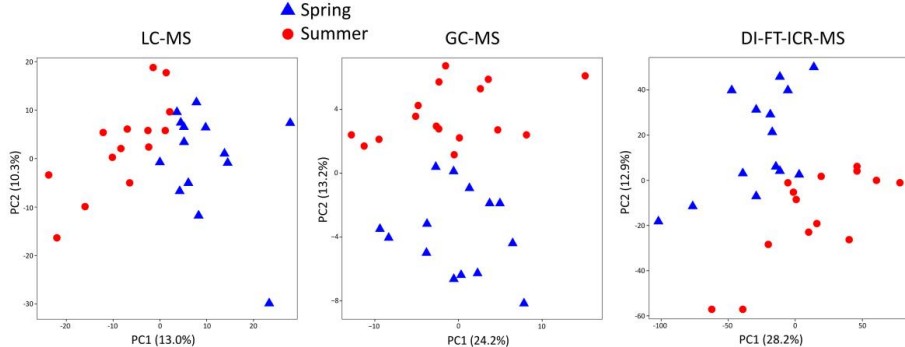



















**Figure 5.** Heat maps of the assigned metabolomic data from the fingerprints obtained from LC-
MS Orbitrap (LC-MS) (a) and GC-MS single quadrupole (GC-MS) (b) for the two sampled
seasons (spring and summer). The colors represent the relative abundance of the metabolite
between seasons. Red represents the highest relative abundance. Metabolites marked by an
asterisk or a cross presented differences ($P < 0.05$) or marginally significant differences ($P < 0.1$)
between seasons after t-test.

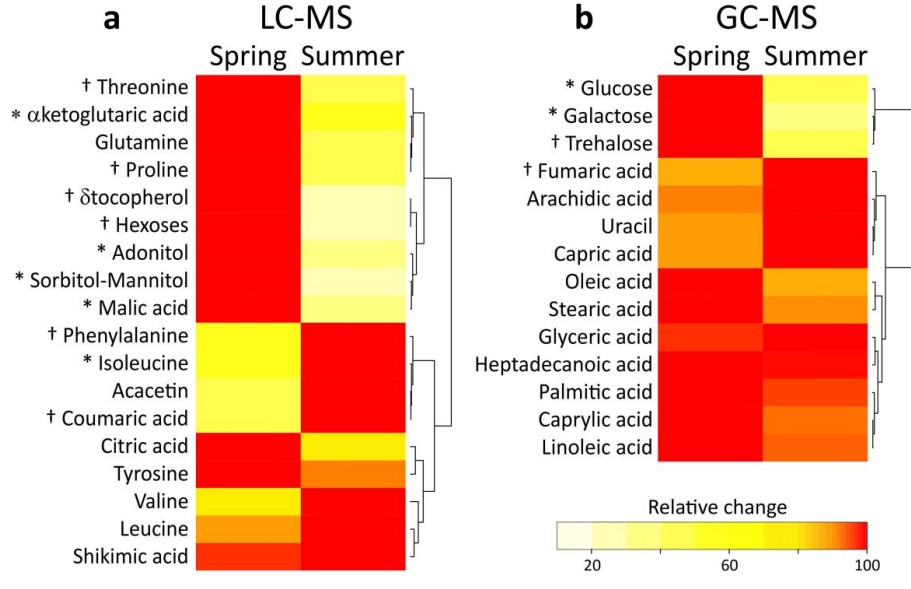














**Figure 6**. Box plots for the proportion (%) of the CHNO, CHO and CHOS (a) and CHNOS, CHNOP,
CHOSP, CHNOSP and CHOP (b) formula classes for spring and summer. Box plots show median
values of each feature. Extreme values are shown in open dots. Asterisks denote statistical
significance between spring and summer for each comparison ($P < 0.05$ (*); $P < 0.0001$ (****)),
and black dots denote marginal significance ($P < 0.1$).

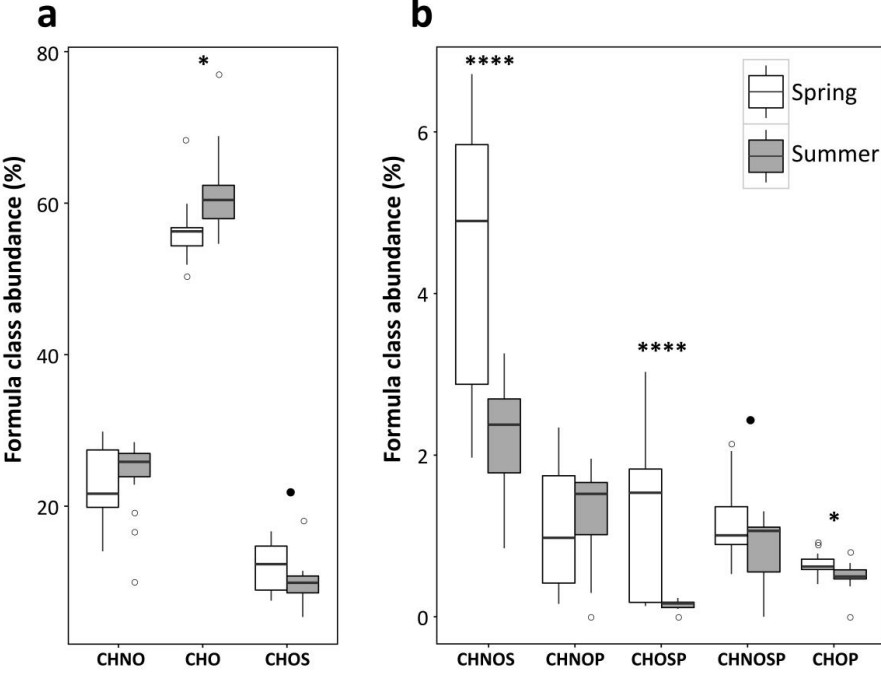













**Figure 7.** van Krevelen (vK) diagrams plotted with DI-FT-ICR data. (a) VK of the unique assigned
features observed spring (blue) and summer samples (red). Classic compound classification
areas are shown in the diagram. (b) vK diagram of all the assigned features represented by the
relative intensity of spring relative to summer. Darker blue dots represent higher relative
intensity in spring and darker red dots represent higher relative intensity in summer. The three
different areas drawn in the vK are regions with higher feature relative intensity in spring or
summer.

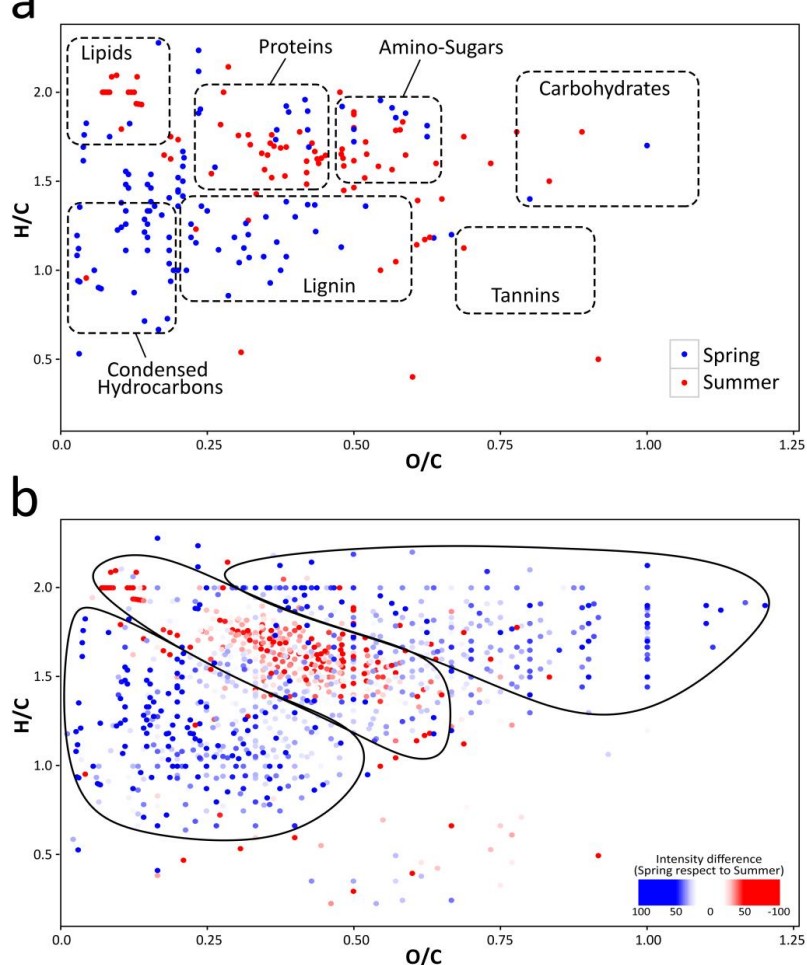







**Figure 8.** Carbon number versus m/z (CvM) (a) and Oxygen/Carbon ratio versus mass (b)
diagrams preformed with DI-FT-ICR data and represented by the relative intensity of spring
relative to summer. Darker blue dots represent higher relative intensity in spring and darker
red dots represent higher relative intensity in summer. Mean (±SE) of Oxygen/Carbon of the
features detected in spring and summer aerosols (c). Statistic-t and P values are shown in the
graph.

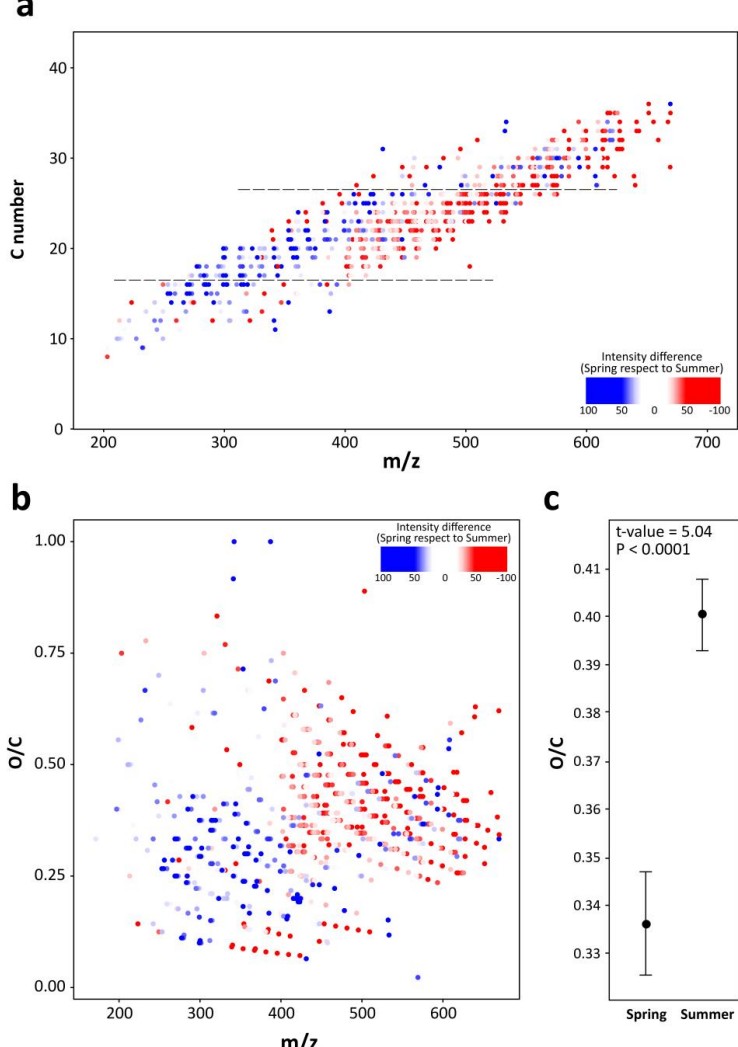

