# Peer review of "Atmo-metabolomics: a new measurement approach for investigating aerosol composition"

_Atmospheric Measurement Techniques, 2016_

## Referee Comment (RC1) · Anonymous Referee #2 · 2 Oct 2016

Overview: This manuscript describes a metabolic-approach for the analysis of atmospheric aerosol. The approach includes GC/MS, LC/MS and direct injection FT-ICR-MS measurements. To demonstrate the potential for this method to contribute toward an improved understanding of natural metabolites associated with aerosol, the authors studied the composition of aerosol collected in the spring and the summer. Key results include: the finding that plant-related metabolites (namely organic acids and carbohydrates) are higher in the spring than summer; the summer samples included metabolites associated with oxidative stress; and summer aerosol composition included a higher fraction of high molecular weight compounds than spring with a higher O/C ratio.

[Figure]

The manuscript contains very valuable laboratory method information that is well-referenced. However, the details about the advanced statistical analysis are deficient. The introduction and methods sections are well-written, but the results and discussion section seems to be presented poorly. Given the inadequate description of the statistical approach, I found the results section to be especially difficult to understand. Another aspect for further consideration is placing this work into the context of the current literature on aerosol chemistry. There's quite a bit of similar work without a so-called "metabolomics" approach that is relevant.

Specific suggestions: * The literature review of atmospheric aerosol composition is weak and outdated. Since the authors claim to be the first to apply metabolomics techniques to aerosol, which are not necessarily different from other composition measurements, it would be nice if they would acknowledge the vast literature of GC/MS, LC/MS and FT-ICR-MS results aimed at understanding aerosol composition. * Lines 102 - 106: How important is the carbon and nutrient deposition of aerosols to ecological systems? * Lines 145 - 148: The atmospheric system is quite complex and the goals of this manuscript are quite broad. I suggest some refinement of the manuscript goals with a focus on a well-defined portion of the atmospheric system, since this work doesn't address larger spatial sampling, research flight measurements, or multiphase measurements. * Line 187: I often see this statement in manuscripts, but it is not a realistic resolving power for environmental samples. Can the authors cite a paper demonstrating the successful measurement of a complex mixture with a resolving power and actual resolution of 1,000,000? * The organization of sections 2.3 - 2.5 is a little bit strange. Specifically, a description of the GC/MS sample prep (in 2.3) is given followed by LC/MS analysis (2.4), which is in turn followed by the GC/MS analysis (2.5). * Line 346: How were both positive and negative ionization performed with LC/MS? Were they done in separate runs or using fast polarity switching? * Line 371: Was negative mode ESI performed? Why was negative ESI not performed for atmospheric aerosol characterization? * Lines 381 - 383: Both fragment ions and exact mass were used to assign metabolites. Were these measurements made in single runs LTQ MS/MS and FT-MS

in tandem or something else? * Line 418: Why was S/N > 7 used as a threshold? How was the S/N determined? * Lines 473 - 476: How many data points were used for this analysis? How were the sub- sets of data selected for analysis? Some discussion on the QA filtering procedures and selection of data for statistical analysis is greatly needed. * Line 487: In what sense is the statistical significance? * Lines 489-496: What do these compounds indicate? How were they identified? * Lines 501-504: This approach from Kim et al. is highly speculative. It's also not an appropriate approach for atmospheric aerosol. Did you extra proteins? How did you verify protein-like components? * Lines 517 - 520: What is the meaning of this observation? * Aerosol sampling information is vague and seems to imply that the authors are unfamiliar with standard sampling techniques for atmospheric chemistry. How did you assess the total carbon concentrations, filter artifacts, and other recovery issues? * Sampling flow rates are expected to change with diurnal cycles (e.g., temperature & pressure); how was this recorded or accounted for? * Lines 535 - 537: The purpose of the study was to assess the sensitivity of different mass spectrometry instruments. But, I didn't understand how that was accomplished? Did you define method detection limits or find any limitations in your approach? More discussion on this would be appreciated. * How does you approach differ from the existing approaches to canopy measurements or other ecological studies focused on atmospheric-biosphere exchange? * Lines 584-587: Which solvents did you use to sequentially extract the filters? How did you evaluate the results of various solvent combinations? * Lines 590-591: What was quantified in your study? * Lines 596 - 600: How was the absorption extract recovery assessed? * Line 623: "match" or assign? * Lines 706-710: Please clarify how the "metabolic fingerprint" was defined/classified? * Table 1: Fingerprint information is unclear. Please add some explanation in the body of the paper. * Figure 1: What about aqueous phase processing of VOCs or aerosol? * Figure 3: How were common inorganic ions removed from the samples before Di-FT- ICR-MS? * Figure 5: I assume this is the list of "metabolic fingerprint" species. Please clarify. * Figure 7: How were the species in (a) subsetted from the whole dataset?

---

## Referee Comment (RC2) · Anonymous Referee #1 · 7 Oct 2016

Referee report for "Atmo-metabolomics: a new measurement approach for investigating aerosol composition and ecosystem functioning" by Albert Rivas-Ubach et al., submitted to AMT

This manuscript describes the organic analysis of ambient aerosols with three techniques, GC-MS, LC-MS and direct infusion MS. The focus of this manuscript is not clear at all. The title seems to suggest that a new technique is described but all that is provided are analyses techniques that are used in the community since years. So, I cannot see what the new aspect of this paper is. Creating a new word for existing analysis strategies is not helpful.

It is not clear what the focus of the paper should be. The actual results seem to suggest

that tracers of PBAP are the main focus but then the manuscript often mentions that the aim is to determine the overall particle composition, which is clearly dominated by many other sources ad not only PBAP.

Figure 1-3 are to a large extent trivial and would be better suited in a review rather than in a research paper.

p. 4 looks to me more like a conclusion section rather than text for an introduction.

The results (e.g. in Figure 4-8) show some interesting findings but overall they are hardly discussed and compared to existing, up to date literature. For all the applied techniques (GC, LC, and direct infusion high resolution MS) there are many current publications, which need to be discussed.

Aerosol sampling (p. 8). It is not clear why the commercial filter holders were modified. This should be clearly motivated. Filter sampling is used in aerosol sciences since decades and it is a standard method. However, much of the sampling description seems to suggest that a new technique is presented, which is not the case. Aerosol was collected without any upper size cut as it is standard practice in aerosol science. This is a serious short-coming and brings the severe risk that large biological material is collected that would not be transported over significant distances due their large size. Collecting aerosol within a certain size range is absolutely essential for any aerosol sampling and analysis. Therefore, the results of this study cannot claim to represent atmospheric aerosols.

GC and LC results. LC results report 18 identified compounds. GC analysis mention 14 compounds. Most comprehensive aerosol analyses presented in the literature using these techniques identify many more compounds. It is not clear why in the study presented here only a small number of compounds was identified. There is no evidence given how these compounds were identified. Simply mentioning "Library identification" is not sufficient. More details would need to be given.

Figure 7 and 8 show interesting results but more discussion would be needed.

Section 4.1 is mostly trivial discussion and can be shortened a lot. The same applies to much of section 4.3.

---

## Author Comment (AC1) · 30 Nov 2016

To editor and reviewers, We thank very much the reviewers for their effort and their time to read our manuscript and provide very valuable comments. After carefully reading all the comments of the two referees, we understand that our manuscript could create confusion. Our main intention was to describe an approach to advance ecological research with all the necessary steps, including sampling, extraction of metabolites in liquid phase, analyses with 3 different instruments, data mining and analysis, to obtain the metabolomic fingerprints from particles in suspension in the lower atmosphere. We expect this method will be useful for addressing novel questions in ecology and other related disciplines. Therefore, it is a methodologic manuscript oriented especially to ecological research and is not intended to target atmospheric chemistry studies, although we recognize that this is more of the focus for the AMT journal. Since this method describes all the details necessary to characterize the metabolomes of aerosols, we thought that AMT was a suitable target journal. However, while we think that our methodology does provide very valuable information for atmospheric scientists, we recognize that this method is aimed mainly to assist the ecological community and now acknowledge that the scope of AMT is more oriented to publish research focused in "advances in remote sensing, as well as in situ and laboratory measurement techniques for the constituents and properties of the Earth's atmosphere". After carefully reading all the reviewer comments we believe that our manuscript would probably fit better to an ecological journal but since our manuscript was still considered for revision and will be published in the online discussion, we have now modified the text clarifying the aims of the method. Note that it does address any of the issues related to the field of aerosol chemistry, as this was never our intention. We hope the aims of this methodological manuscript are now clearer.

Anonymous Referee #1 Referee report for "Atmo-metabolomics: a new measurement approach for investigating aerosol composition and ecosystem functioning" by Albert Rivas-Ubach et al., submitted to AMT This manuscript describes the organic analysis of ambient aerosols with three techniques, GC-MS, LC-MS and direct infusion MS. The focus of this manuscript is not clear at all. The title seems to suggest that a new technique is described but all that is provided are analyses techniques that are used in the community since years. So, I cannot see what the new aspect of this paper is. Creating a new word for existing analysis strategies is not helpful. It is not clear what the focus of the paper should be. The actual results seem to suggest that tracers of PBAP are the main focus but then the manuscript often mentions that the aim is to determine the overall particle composition, which is clearly dominated by many other sources ad not only PBAP.

Response: As mentioned previously, our manuscript is focused on the ecological community. While these techniques have been available to the scientific community, these metabolomic techniques have not been used to characterize the metabolomes of aerosols. The aim is to provide ecologists and environmental scientists with a tool to assess the ecosystem status and stress levels through the atmospheric detection of biochemical compounds. To demonstrate our method, we collected aerosol samples from two distinct seasons to test our methodology but without attempting a deep analysis of the differences between the seasons. We simply chose seasonality for comparison purposes and to test the sensitivity of the instruments to detect differences between the metabolomic fingerprints; other factors such as different ecosystems could have been chosen too. Metabolomics techniques, which include all the steps from the sampling to the analysis of the data, have been widely used to measure the metabolomes from living systems. However, metabolomics can also be applied to obtain metabolic signatures from any sample contaning natural organic matter (NOM). We acknowledge that diverse mass spectrometry techniques such as GC-MS have been used for years in the atmospheric research, especially to detect and quantify volatile species such as BVOCs. Nonetheless, our purpose in this manuscript was never to improve or replace those well-defined approaches. With the approach that we present in this manuscript, researchers should be able to detect metabolites in aerosols directly linked with the main physiological processes occurring in living organisms. Moreover, our manuscript provides a good synthesis of the main techniques used for metabolomic analyses, including FT-ICR-MS, which we believe it is especially useful for those researchers interested to introduce metabolomics approaches to further understand the link between the atmosphere and ecosystems

Figure 1-3 are to a large extent trivial and would be better suited in a review rather than in a research paper.

Response: The main intention of our manuscript was to explain in detail a methodology of sampling particles in suspension. Figure 1 shows the most common sources of compounds and particulates as well as their major roles and their interaction with ecosystems, humans and climate. We think it is useful to provide a general background, especially to the readers from ecological and other environmental disciplines, of the main sources and processes of compounds in the atmosphere. Figure 2 describes the sampling method; we consider this to be important for a methodological paper; however, following the referee's comment, we have now moved this figure to the supporting information (Figure S1). This figure provides a general picture of how the sampling was performed and how the cassettes should hold the filters for a homogeneous sampling; important information for researchers that are not familiar with aerosol sampling. Figure 3 (now Figure 2) describes step by step how to extract the metabolites from the filters into a solution. We think showing this figure in the main text of a methodological article for metabolomics analyses is necessary.

p. 4 looks to me more like a conclusion section rather than text for an introduction. The prupose of this section is to make the case for why this approach would be useful for ecologists and other disciplines. It is not based on the results of our study but just shows the need and the potential value of demonstrating this approach.

Response: The results (e.g. in Figure 4-8) show some interesting findings but overall they are hardly discussed and compared to existing, up to date literature. For all the applied techniques (GC, LC, and direct infusion high resolution MS) there are many current publications, which need to be discussed. The main objective of our methodological article is to provide the proof that the "atmo-metabolomes" (metabolomic fingerprints) of spring and summer differ statistically between them. Additionally, the filters were analyzed with three different instruments (GC-MS, LC-MS, FTICR) to test whether they were sensitive enough to detect significant changes ($P < 0.05$) between seasons. To discuss all the details obtained from each instrument would shift the aim of this manuscript and it would considerably lengthen the text. One of the main aims of the study was to provide a method able to discern the differences in aerosol metabolomes between two different seasons with three different instruments. We acknowledge the large aerosol bibliography available using the MS techniques. However, as discussed above, this article is not intended to be a review of all the bibliography or a revision of current atmospheric sampling techniques. We agree that it is important to show a certain properly referenced background directly related with our instruments and results. In the new version of the manuscript we restructured the introduction providing more information related to MS studies. The space in a journal is limited and the section providing discussion of the results from the three mass spectrometry instruments, which is not the central focus of the article, is already almost 800 words. For this reason, we have not extended this part in order to keep the aim of the study clear in a relatively concise manuscript.

Aerosol sampling (p. 8). It is not clear why the commercial filter holders were modified. This should be clearly motivated.

Response: Filter cassettes do not require modification if they already ensure a homogeneous distribution of the air-flow along the filter surface during the sampling. The commercial cassettes we used in our study were designed in a way that the air did not flow properly along the entire surface of the filters, so they had to be slightly modified to achieve that homogeneous distribution of the air-flow. We wanted to provide all the sampling details in the manuscript but we finally decided to delete those sentences from the M&M section to avoid any confusion.

Filter sampling is used in aerosol sciences since decades and it is a standard method. However, much of the sampling description seems to suggest that a new technique is presented, which is not the case. Aerosol was collected without any upper size cut as it is standard practice in aerosol science. This is a serious short-coming and brings the severe risk that large biological material is collected that would not be transported over significant distances due their large size. Collecting aerosol within a certain size range is absolutely essential for any aerosol sampling and analysis. Therefore, the results of this study cannot claim to represent atmospheric aerosols.

Response: We agree that our sampling procedures do not differ substantially from the ones used in atmospheric sciences. As a methodological article, its aim is to explain in detail all the main steps in order to obtain the metabolomic fingerprints of the particles in suspension in the low atmosphere with different analytical instruments. In this article we put together in a comprehensive way, especially for the ecological and plant science community, all those steps and we are convinced that a detailed description of our sampling method is necessary. However, contrary to many aerosol sampling methods, our methodology is very flexible, portable and economic, and probably the most important aspect: very simple. As mentioned above, we wanted to put all the details together in a single manuscript and we expect it to be valuable for the research community. Furthermore, as mentioned by the referee, our sampling method can collect large biological material. While an upper cut size can be employed to increase the footprint of the ecosystem represented, the approach used here is suitable for characterizing the ecosystem of the immediate surrounding area which was our objective for this study. As stated before, the aim of this article was not to provide answers on the chemistry processes occurring in the atmosphere in a specific moment but to explain a method to obtain the metabolomic profiles of the particles in suspension in the atmosphere. For this reason, it is not necessary to sample a specific size range but that sample can include all particles to obtain a general picture of which molecular compounds are present in the particle fraction in the lower atmosphere. We are convinced that this methodological approach provides very valuable information for the ecological community.

GC and LC results. LC results report 18 identified compounds. GC analysis mention 14 compounds. Most comprehensive aerosol analyses presented in the literature using these techniques identify many more compounds. It is not clear why in the study presented here only a small number of compounds was identified. There is no evidence given how these compounds were identified. Simply mentioning "Library identification" is not sufficient. More details would need to be given.

Response: The number of compounds identified and verified in a sample depends mainly on three factors: i) the solvents used for the extraction of metabolites, ii) the concentration of metabolites in the samples and iii) the specific metabolites present in the metabolite databases. Typically, un-targeted metabolomics techniques have been applied to obtain the metabolic fingerprints and profiles from living organisms. For this reason the metabolite databases include metabolites from living organisms and mainly from their primary metabolism. The focus of our method are compounds known to be metabolites coming from living organisms. According to the metabolite databases used for this study we assigned a bit more than 30 compounds combining both GC and LC-MS methodologies that are directly linked to the metabolism of organisms, likely from plants. Additionally, sampling was performed in an area with very low biological activity compared to more forested areas and organic volatile compounds derived from plants or anthropogenic emissions could not be identified by our libraries since those compounds are simply not normally listed in our metabolomic libraries. As we mentioned in the manuscript: "The techniques to characterize the gas phase component of atmo-metabolomes are well described elsewhere (Smith and Španěl, 2011; Tholl et al., 2006). Our purpose here is to describe an atmo-metabolomic method for sampling aerosols and characterize the particle phase of the atmo-metabolomes.". In the past version of the manuscript we also mentioned that "Metabolite assignation with LC-MS was performed by our metabolite library with more than 200 typical metabolites usually present in plants and fungi including products from primary and secondary metabolism"; so we made it clear which kind of compounds we were targeting. Our intention was never to reproduce a method to sample all atmospheric organic compounds but to measure with LC-MS and GC-MS compounds present in solid particles coming from living systems. We have rewritten the section regarding the metabolite identification by LC-MS. It now reads: "Metabolite assignation with LC-MS was performed by our metabolite library with more than 200 typical metabolites usually present in plants and fungi including products from primary and secondary metabolism. Assignation were performed separately for each ionization mode (positive and negative) and using the exact mass of metabolites and RT. ESI do not typically fragment all ions, however, in some molecules we could still detect some fragments which were also considered for the metabolite assignation and relative quantification. According to Sumner et al., 2007, our LC-MS metabolite assignment is putative since it was based on total exact mass of the metabolite and RT of standard measurements in the instrument. However, the use of high MS resolution achieved with Orbitrap technology and RT reduces substantially the number of false positive assignations. For more detailed information regarding the metabolite assignation see Rivas-Ubach et al., (2016b). RT and m/z values of metabolite matching for LC-MS are shown in Table S2."

Figure 7 and 8 show interesting results but more discussion would be needed.

Response: Already responded above.

Section 4.1 is mostly trivial discussion and can be shortened a lot. The same applies to much of section 4.3.

Response: Following the referee's advice, we have now shortened the section 4.1 and deleted the section 4.3 re-organizing some content into the introduction.

Anonymous Referee #2

Overview: This manuscript describes a metabolic-approach for the analysis of atmospheric aerosol. The approach includes GC/MS, LC/MS and direct injection FT-ICRMS measurements. To demonstrate the potential for this method to contribute toward an improved understanding of natural metabolites associated with aerosol, the authors studied the composition of aerosol collected in the spring and the summer. Key results include: the finding that plant-related metabolites (namely organic acids and carbohydrates) are higher in the spring than summer; the summer samples included metabolites associated with oxidative stress; and summer aerosol composition included a higher fraction of high molecular weight compounds than spring with a higher O/C ratio. The manuscript contains very valuable laboratory method information that is well referenced. However, the details about the advanced statistical analysis are deficient. The introduction and methods sections are well-written, but the results and discussion section seems to be presented poorly. Given the inadequate description of the statistical approach, I found the results section to be especially difficult to understand. Another aspect for further consideration is placing this work into the context of the current literature on aerosol chemistry. There's quite a bit of similar work without a so-called "metabolomics" approach that is relevant.

Response: Many thanks for the positive evaluation on the interest of the study and for considering very valuable the laboratory method information. We have now rewritten the introduction, focusing it on the ecological applications of the study of the metabolomic fingerprint of ecosystems on atmospheric aerosols. And to address the referee's concerns, we have now clarified the statistical methods section and combined the results and discussion. We hope that now the text is clearer. We acknowledge that GC-MS and other mass spectrometry techniques have been widely used in the atmospheric research. Nonetheless, as in our response to the previous referee, our purpose for this manuscript was not to improve or replace those well-defined approaches or to investigate the chemistry of the atmosphere. We present an approach that is novel and useful for the ecological community by enabling researchers to detect aerosol metabolites that may be directly linked with the main physiological and ecological processes of living organisms.

Specific suggestions:

The literature review of atmospheric aerosol composition is weak and outdated. Since the authors claim to be the first to apply metabolomics techniques to aerosol, which are not necessarily different from other composition measurements, it would be nice if they would acknowledge the vast literature of GC/MS, LC/MS and FT-ICR-MS results aimed at understanding aerosol composition.

Response: Our manuscript aims to describe in enough detail the set of necessary procedures to obtain the metabolome profiles from aerosols. As mentioned above, this method is mainly focused to detect signatures directly linked to the main physiological and ecological processes of organisms; metabolites which are not volatile but are also part of many particles in suspension in the atmosphere. We have modified the introduction more clearly focusing the aims on the ecological aspects.

Lines 102 - 106: How important is the carbon and nutrient deposition of aerosols to ecological systems?

Response: We have expanded the section in the introduction about the aerosol deposition on ecosystems.

Lines 145 - 148: The atmospheric system is quite complex and the goals of this manuscript are quite broad. I suggest some refinement of the manuscript goals with a focus on a well-defined portion of the atmospheric system, since this work doesn't address larger spatial sampling, research flight measurements, or multiphase measurements.

Response: As mentioned above, we have rewritten the introduction section. We have now better focused our manuscript and hope the purposes of our method are clearer.

Line 187: I often see this statement in manuscripts, but it is not a realistic resolving power for environmental samples. Can the authors cite a paper demonstrating the successful measurement of a complex mixture with a resolving power and actual resolution of 1,000,000?

Response: We reviewed the capability of FT-ICR-MS in the manuscript as an introduction of this analytical method. Therefore, we have to report the maximum resolving power that FT-ICR-MS can achieve. However, we did not state that such resolving power is currently used in environmental study. We stated the actual resolving power ($\sim$400,000 at 400 m/z) for our samples.

The organization of sections 2.3 - 2.5 is a little bit strange. Specifically, a description of the GC/MS sample prep (in 2.3) is given followed by LC/MS analysis (2.4), which is in turn followed by the GC/MS analysis (2.5).

Response: We understand that this may create some confusion, however, we wanted to be consistent and we have followed the same order for the methods and results along the article; LC-MS, GC-MS and FTICR consecutively. The section 2.3 described the extraction of metabolites from the quartz filters which is common for all the three MS techniques (LC, GC and ICR). However, differently to LC-MS and DI-FT-ICR extracts, samples for GC-MS require an additional step; the derivatization of metabolites. This step is also indicated in the Figure 3 and it is clearly linked to the extraction of metabolites. We considered the derivatization should not be in the following section of GC-MS analyses (2.5). However, we could consider moving this section if required. After the section for sample preparation (2.3)(common for the three techniques), we have described the parameters used for each one of the MS instruments separately according to the order established (2.4 for LC-MS, 2.5 for GC-MS and 2.6 for DI-FTICR). Following the instrument analysis sections, the next 3 sections (2.7, 2.8 and 2.9) provide the details to obtain the numerical data from each of the instruments. Also, these 3 sections follow the same order established, so 2.7 for LC-MS, 2.8 for GC-MS and 2.9 for FT-ICR.

So, our logic for the description of the methods was: 1. Extraction of metabolites. (2.3) 2. Data acquisition by each MS instrument (2.4, 2.5 and 2.6) 3. Processing of MS chromatograms/spectra from each instrument. (2.7, 2.8 and 2.9) We think that this order is comprehensive; however, we can change the distribution of the methods if required. An option would be to include all the instruments in a single "Data acquisition" and "Processing of chromatograms/spectra" section by using subtitles.

Line 346: How were both positive and negative ionization performed with LC/MS? Were they done in separate runs or using fast polarity switching? The LTQ Orbitrap Velos cannot switch ionization polarities quickly. Only the most recent Q-Exactive and the new LUMOS Orbitrap versions can operate with fast polarity switch. So samples were first injected in positive mode and then in negative mode. We now have indicated this detail in the manuscript and it can be read as: "All samples were first analyzed in positive (+) ionization mode and later in negative (-) ionization mode."

Line 371: Was negative mode ESI performed? Why was negative ESI not performed for atmospheric aerosol characterization?

Response: Analyses in FT-ICR-MS were performed exclusively in negative mode as already mentioned in the manuscript: "Samples were directly infused into the mass spectrometer using a standard Bruker electrospray ionization (ESI) in negative mode at a flow rate of 3.0 $\mu$L/min through an Agilent 1200 series pump (Agilent Technologies, Santa Clara, CA, USA ." FT-ICR-MS in negative mode is the most used method to investigate natural organic matter. While positive ESI mode could increase the compound coverage, we opted to use negative mode only as our instrument was optimized under ESI(-) for organic matter exploration.

Lines 381 - 383: Both fragment ions and exact mass were used to assign metabolites. Were these measurements made in single runs LTQ MS/MS and FT-MS in tandem or something else?

Response: Although we already referenced a manuscript where the metabolite assignation is well described, we agree that this section should be more detailed, especially for a methodological article like the present one. We have extended this section of the manuscript. It now reads: "Metabolite assignation with LC-MS was performed by our metabolite library with more than 200 typical metabolites usually present in plants and fungi including products from primary and secondary metabolism. Assignation were performed separately for each ionization mode (positive and negative) and using the exact mass of metabolites and RT. ESI do not typically fragment all ions, however, in some molecules we could still detect some fragments which were also considered for the metabolite assignation and relative quantification. According to Sumner et al., 2007, our LC-MS metabolite assignment is putative since it was based on total exact mass of the metabolite and RT of standard measurements in the instrument. However, the use of high MS resolution achieved with Orbitrap technology and RT reduces substantially the number of false positive assignations. For more detailed information regarding the metabolite assignation see Rivas-Ubach et al., (2016b). RT and m/z values of metabolite matching for LC-MS are shown in Table S2."

Line 418: Why was S/N > 7 used as a threshold? How was the S/N determined?

Response: S/N was determined in the Bruker Data Analysis software, which was assessed based on baselines near each peak. S/N of 3 and 5 are often used in natural organic matter exploration as that range is considered as the mínimum detection limit (Riedel and Dittmar 2014). We chose S/N>7 for a more conservative measure.

Lines 473 - 476: How many data points were used for this analysis? How were the subsets of data selected for analysis? Some discussion on the QA filtering procedures and selection of data for statistical analysis is greatly needed.

Response: Each analytical technique generated their own data that were posteriorly analyzed separately. All the data (metabolomic fingerprints) from each instrument were used to perform the PERMANOVAs. PERMANOVAs were performed separately. As this manuscript was not especially focused on the understanding of the metabolomes or chemical signatures between summer and spring aerosols, we did not include the number of features we observed and used for the statistical analyses. We can include this information if you think it necessary. We have added some more text in the material and methods section explaining the data filtering in more detail. Now the text reads: "For each season (spring and summer) and dataset (LC-MS, GC-MS and FT-ICR-MS), the variables present in less than 50% of the samples were excluded for the statistical analyses. The signal values measured in the experimental blanks in each of the instruments were subtracted from the datasets. Each of the variables from metabolome fingerprints obtained from each MS instrument were posteriorly submitted to Levene's and Shapiro tests to assess homogeneity of variances and normality, respectively. Variables that did not comply with those statistical assumptions were removed from the datasets. Outlier measurements were replaced for missing values and were defined as those measurements of a specific variable with values three-fold higher than the third quartile or three-fold lower than the first quartile of each season. For FT-ICR-MS datasets we have been very conservative and only the formula assigned features that presented less than 0.3ppm of error were used although cutoff values up to 0.5ppm are typically used (Osterholz et al., 2016)."

Line 487: In what sense is the statistical significance?

Response: As typically used in the vast majority of environmental studies, the alpha error or type I error is maintained at 5%. The term "statistical significance" is widely used for P values lower than 0.05 for a given test. So, alpha error (type I error), the probability of rejecting the null hypothesis when is true, was maintained at 5%.

Lines 489-496: What do these compounds indicate? How were they identified?

Response: In the results section we only indicate which compounds increased significantly (P <0.05) or marginally significantly (P <0.1) in the spring samples. Some of the metabolites identified are briefly discussed in the discussion section (4.4). We did not discuss all the results obtained with each of the instruments since it would be out of the main aim of the study. This article is just a methodological article and we have focused the discussion on the major results and it was not our intention to investigate all of the differences between the seasons. Those compounds were identified according to our LC-MS database of metabolites of plants and fungi, however, as already mentioned above we have now extended the section of metabolite identification and provided more details.

501-504: This approach from Kim et al. is highly speculative. It's also not an appropriate approach for atmospheric aerosol. Did you extra proteins? How did you verify protein-like components?

Response: We highly agree with the referee. Although the compound classification obtained from van Krevelen (vK) diagrams (O:C vs. H:C) provides a certain approximation of the composition of the samples, we also think that their use should be limited. However, vK diagrams are widely used to understand the chemical changes in samples and this classification is still widely used to represent the FTICR data. Because this compound classification is a widely used method to understand organic matter composition, our intention was to show this to the readers. However, it should be noted that even in the previous manuscript version we only briefly mentioned this classification. In fact, we are already working on another manuscript reviewing this commonly used compound classification for FTICR data. For this reason, in the new version of the manuscript we finally decided to retain the review of the existence of such classification but we have deleted the previous Figure 7.

Lines 517 - 520: What is the meaning of this observation?

Response: Here we mention that particles in summer showed significantly higher intensities in features with higher O/C ratios. This result is briefly discussed in the discussion method, however, as the aim of the article is solely methodological, we did not discuss each of the results in depth. We simply chose two seasons to test if we could detect statistically significant differences between the "atmo-metabolomes" between the two seasons. Different factors could be chosen for this test, like two different ecosystems but we considered that seasonality was more a feasible and comprehensive factor to test.

Aerosol sampling information is vague and seems to imply that the authors are unfamiliar with standard sampling techniques for atmospheric chemistry. How did you assess the total carbon concentrations, filter artifacts, and other recovery issues?

Response: Our intention was not to reproduce a standard atmospheric chemistry sampling technique as we recognize that there are numerous researchers focused on the chemistry transformations in the atmosphere and for that reason many specific protocols are typically used. However, our simple method is suitable for characterizing the metabolome of the atmosphere. The aim of metabolomics is to compare relatively different groups of samples. Since it is practically impossible to obtain a full metabolome in terms of absolute concentrations for each of the detected metabolites, as long as the sample preparation is performed equally for all the samples we can perform a relative comparison between groups of samples. Filter artifacts were coped with experimental blanks that were injected to all instruments and any signal obtained from those blanks was posteriorly subtracted from the original samples. The use of blanks is a standard procedure for any metabolomics study. We have now included more details in the material and methods to respond to those concerns.

Sampling flow rates are expected to change with diurnal cycles (e.g., temperature & pressure); how was this recorded or accounted for?

Response: Each filter was sampled exactly for the same amount of time and in the same time range as described in the material and methods section: "The pump was working daily during 18 consecutive hours and pumped air at 30 L per minute through each filter. Filters were replaced manually before 09:00am and the pump started working automatically at 09:00am and stopped automatically at 03:00am the following day. Filters were stored at -80áţŠC until metabolite extraction. Filters were sampled on a tower at 8 meters height."

Lines 535 - 537: The purpose of the study was to assess the sensitivity of different mass spectrometry instruments. But, I didn't understand how that was accomplished? Did you define method detection limits or find any limitations in your approach? More discussion on this would be appreciated.

Response: We rely on the statistical analyses to test the sensitivity of the used techniques to detect changes between seasons. We sampled in an area with a very low primary producer activity and still we were able to detect significant differences in the overall atmo-metabolomes between spring and summer. The significance obtained in the PERMANOVA test proves that each of the techniques was sensitive enough to detect changes between those samples. The principal component analyses (PCAs) for each of the instruments also prove that the instruments were able to detect significantly different overall composition in the spring vs. summer samples. In order to clarify this concern, we have modified the text properly in different sections.

How does you approach differ from the existing approaches to canopy measurements or other ecological studies focused on atmospheric-biosphere exchange?

Response: In this article we explained, and put together, the different steps to obtain the metabolomic fingerpints (or metabolomic signatures) from particles sampled in the lower atmosphere. As far as we know, no other approach for analyzing aerosol metabolomes has been published. Similar sampling methods can be performed in other ways with different pumps and filters, however, the method we propose is more portable (lower weight and volume), flexible (can be easily manipulated in different ways) and more economic than the commonly commercialized prototypes for aerosol filter sampling. Also as a methodological article we provided detailed information on how our sampling was designed and performed. As discussed in the manuscript, the main idea is to obtain the minimum values in the filter-size/pump-flow ratio to concentrate as much as possible the filters. Our objective was not to perform a comparative study with all the available sampling methods. We just described a very simple and flexible method that samples particles in suspension efficiently and at a low cost. In addition, researchers can choose the filter size they require while many commercial systems are compatible only with a unique filter size.

Lines 584-587: Which solvents did you use to sequentially extract the filters? How did you evaluate the results of various solvent combinations?

Response: We did not perform a sequential extraction in this study. We used methanol:water (80:20) as one of the most widely used solvent mix for extraction of metabolites. We cite different studies and methods where the number of extractions and recovery is discussed. We did not attempt to use a whole variety of extraction methods; we only aimed to show a generally used extraction method to investigate whether the analytical techniques can differenciate statistically the metabolomes between spring and summer aerosols. We also mention that this extraction method is not exclusive but suggested and indicate that different extraction methods can be also used. As widely discussed in several analytical chemistry articles, different extraction methods obtain different range of metabolites based mainly on their polarity.

Lines 590-591: What was quantified in your study?

Response: In this study we performed a relative quantification of the metabolomic fingerprints for comparative analyses between spring and summer.

Lines 596 - 600: How was the absorption extract recovery assessed?

Response: We measured how much volume of solvent was recovered (after the extraction procedures) with respect to the initial solvent added. In the text we mention that we can get an extraction recovery of 89% which indicates that we recover 0.89mL per each 1mL added to the tubes with the filters to perform the extraction. We did not think that it was necessary to incorporate this information in the methods section. However, we can introduce the explanation if the referee thinks it is necessary.

Line 623: "match" or assign?

Response: We appreciate you made us notice this, we agree "assign" is more suitable than "match" in this sentence. We have changed the word in the manuscript.

Lines 706-710: Please clarify how the "metabolic fingerprint" was defined/classified?

Response: We have added a clarification of metabolomic fingerprints and metabolomic profiles in the introduction section. Now the text reads: "The first step to characterize a metabolome profile is to obtain the chemical signature of the sample (metabolomic fingerprint) without further molecular identification (Sardans et al., 2011). The identification of specific metabolites can be further obtained by the information present in the metabolomic fingerprints. In this study, we describe the different procedures to obtain the metabolomic fingerprints and identify molecular compounds from aerosols. This atmo-ecometabolomics methodology is a potential tool to shed light in novel questions in ecology, especially for the ecosystem-atmosphere interface."

Table 1: Fingerprint information is unclear. Please add some explanation in the body of the paper.

Respone:See comment above.

Figure 1: What about aqueous phase processing of VOCs or aerosol?

Response: We modified the figure but the different atmospheric VOCs transformations are not presented in detail since it was never our intention to address that issue in this manuscript.

Figure 3: How were common inorganic ions removed from the samples before DI-FT-ICR-MS?

Response: It should be noted that most of the inorganic ions are at much lower mass range than our FTICR-MS analytical window (100-1200 m/z). Thus, unless those ions generate clusters that would interfere with the FT-ICR-MS measurements, such as sodium and chloride, removal of inorganic ions were not necessary. In addition, such a problem is more evident in direct infusion positive ion mode, which was not considered in this study.

Figure 5: I assume this is the list of "metabolic fingerprint" species. Please clarify.

Response: As mentioned before we have included the definition of what a metabolic fingerprint is. The list of metabolites does not represent the entire fingerprint of the different seasons but only the portion that has been identified/assigned. We hope it is now clearer.

Figure 7: How were the species in (a) subsetted from the whole dataset?

Response: As explained before, we did not use a subset of the datasets but the whole amount of detected features. However, we have now deleted this figure from the new manuscript version.

Please also note the supplement to this comment:
http://www.atmos-meas-tech-discuss.net/amt-2016-209/amt-2016-209-AC1-supplement.pdf

**Supplement:**

*To editor and reviewers,*

*We thank very much the reviewers for their effort and their time to read our manuscript and*

*provide very valuable comments. After carefully reading all the comments of the two referees,*

*we understand that our manuscript could create confusion.  Our main intention was to describe*

*an approach to advance ecological research with all the necessary steps, including sampling,*

*extraction of metabolites in liquid phase, analyses with 3 different instruments, data mining*

*and analysis, to obtain the metabolomic fingerprints from particles in suspension in the lower*

*atmosphere. We expect this method will be useful for addressing novel questions in ecology*

*and other related disciplines. Therefore, it is a methodologic manuscript oriented especially to*

*ecological research and is not intended to target atmospheric chemistry studies, although we*

*recognize that this is more of the focus for the AMT journal.  Since this method describes all the*

*details necessary to characterize the metabolomes of aerosols, we thought that AMT was a*

*suitable target journal. However, while we think that our methodology does provide very*

*valuable information for atmospheric scientists, we recognize that this method is aimed mainly*

*to assist the ecological community and now acknowledge that the scope of AMT is more*

*oriented to publish research focused in "advances in remote sensing, as well as in situ and*

*laboratory measurement techniques for the constituents and properties of the Earth's*

*atmosphere". After carefully reading all the reviewer comments we believe that our manuscript*

*would probably fit better to an ecological journal but since our manuscript was still considered*

*for revision and will be published in the online discussion, we have now modified the text*

*clarifying the aims of the method. Note that it does address any of the issues related to the*

*field of aerosol chemistry, as this was never our intention. We hope the aims of this*

*methodological manuscript are now clearer.*

Anonymous Referee #1

 Referee report for

"Atmo-metabolomics: a new measurement approach for investigating aerosol composition and ecosystem functioning" by Albert Rivas-Ubach et al., submitted to AMT This manuscript describes the organic analysis of ambient aerosols with three techniques, GC-MS, LC-MS and direct infusion MS.

The focus of this manuscript is not clear at all. The title seems to suggest that a new technique is described but all that is provided are analyses techniques that are used in the community since years. So, I cannot see what the new aspect of this paper is. Creating a new word for existing analysis strategies is not helpful.

It is not clear what the focus of the paper should be. The actual results seem to suggest that tracers of PBAP are the main focus but then the manuscript often mentions that the aim is to determine the overall particle composition, which is clearly dominated by many other sources ad not only PBAP.

*As mentioned previously, our manuscript is focused on the ecological community. While these techniques have been available to the scientific community, these metabolomic techniques have not been used to characterize the metabolomes of aerosols. The aim is to provide ecologists and environmental scientists with a tool to assess the ecosystem status and stress levels through the atmospheric detection of biochemical compounds. To demonstrate our method, we collected aerosol samples from two distinct seasons to test our methodology but without attempting a deep analysis of the differences between the seasons. We simply chose seasonality for comparison purposes and to test the sensitivity of the instruments to detect differences between the metabolomic fingerprints; other factors such as different ecosystems could have been chosen too.*

*Metabolomics techniques, which include all the steps from the sampling to the analysis of the data, have been widely used to measure the metabolomes from living systems. However, metabolomics can also be applied to obtain metabolic signatures from any sample contaning natural organic matter (NOM). We acknowledge that diverse mass spectrometry techniques such as GC-MS have been used for years in the atmospheric*

*research, especially to detect and quantify volatile species such as BVOCs. Nonetheless,*
*our purpose in this manuscript was never to improve or replace those well-defined*
*approaches. With the approach that we present in this manuscript, researchers should*
*be able to detect metabolites in aerosols directly linked with the main physiological*
*processes occurring in living organisms. Moreover, our manuscript provides a good*
*synthesis of the main techniques used for metabolomic analyses, including FT-ICR-MS,*
*which we believe it is especially useful for those researchers interested to introduce*
*metabolomics approaches to further understand the link between the atmosphere and*
*ecosystems*

Figure 1-3 are to a large extent trivial and would be better suited in a review rather than in a
research paper.

*The main intention of our manuscript was to explain in detail a methodology of*
*sampling particles in suspension.*

*Figure 1 shows the most common sources of compounds and particulates as well as*
*their major roles and their interaction with ecosystems, humans and climate. We think*
*it is useful to provide a general background, especially to the readers from ecological*
*and other environmental disciplines, of the main sources and processes of compounds*
*in the atmosphere.*

*Figure 2 describes the sampling method; we consider this to be important for a*
*methodological paper; however, following the referee's comment, we have now moved*
*this figure to the supporting information (Figure S1). This figure provides a general*
*picture of how the sampling was performed and how the cassettes should hold the*
*filters for a homogeneous sampling; important information for researchers that are not*
*familiar with aerosol sampling.*

*Figure 3 (now Figure 2) describes step by step how to extract the metabolites from the*
*filters into a solution. We think showing this figure in the main text of a methodological*
*article for metabolomics analyses is necessary.*

p. 4 looks to me more like a conclusion section rather than text for an introduction.

*The prupose of this section is to make the case for why this approach would be useful*

*for ecologists and other disciplines. It is not based on the results of our study but just*

*shows the need and the potential value of demonstrating this approach.*

The results (e.g. in Figure 4-8) show some interesting findings but overall they are hardly discussed and compared to existing, up to date literature. For all the applied techniques (GC,

LC, and direct infusion high resolution MS) there are many current publications, which need to be discussed.

*The main objective of our methodological article is to provide the proof that the "atmo-*

*metabolomes" (metabolomic fingerprints) of spring and summer differ statistically*

*between them. Additionally, the filters were analyzed with three different instruments*

*(GC-MS, LC-MS, FTICR) to test whether they were sensitive enough to detect significant*

*changes (P < 0.05) between seasons. To discuss all the details obtained from each*

*instrument would shift the aim of this manuscript and it would considerably lengthen*

*the text. One of the main aims of the study was to provide a method able to discern the*

*differences in aerosol metabolomes between two different seasons with three different*

*instruments.*

*We acknowledge the large aerosol bibliography available using the MS techniques.*

*However, as discussed above, this article is not intended to be a review of all the*

*bibliography or a revision of current atmospheric sampling techniques. We agree that*

*it is important to show a certain properly referenced background directly related with*

*our instruments and results. In the new version of the manuscript we restructured the*

*introduction providing more information related to MS studies.*

*The space in a journal is limited and the section providing discussion of the results from*

*the three mass spectrometry instruments, which is not the central focus of the article,*

*is already almost 800 words. For this reason, we have not extended this part in order to*

*keep the aim of the study clear in a relatively concise manuscript.*

Aerosol sampling (p. 8). It is not clear why the commercial filter holders were modified. This should be clearly motivated.

*Filter cassettes do not require modification if they already ensure a homogeneous*

*distribution of the air-flow along the filter surface during the sampling. The commercial*

*cassettes we used in our study were designed in a way that the air did not flow*

*properly along the entire surface of the filters, so they had to be slightly modified to*

*achieve that homogeneous distribution of the air-flow. We wanted to provide all the*

*sampling details in the manuscript but we finally decided to delete those sentences*

*from the M&M section to avoid any confusion.*

Filter sampling is used in aerosol sciences since decades and it is a standard method. However, much of the sampling description seems to suggest that a new technique is presented, which is not the case. Aerosol was collected without any upper size cut as it is standard practice in aerosol science. This is a serious short-coming and brings the severe risk that large biological material is collected that would not be transported over significant distances due their large size. Collecting aerosol within a certain size range is absolutely essential for any aerosol sampling and analysis. Therefore, the results of this study cannot claim to represent atmospheric aerosols.

*We agree that our sampling procedures do not differ substantially from the ones used*

*in atmospheric sciences. As a methodological article, its aim is to explain in detail all*

*the main steps in order to obtain the metabolomic fingerprints of the particles in*

*suspension in the low atmosphere with different analytical instruments. In this article*

*we put together in a comprehensive way, especially for the ecological and plant science*

*community, all those steps and we are convinced that a detailed description of our*

*sampling method is necessary. However, contrary to many aerosol sampling methods,*

*our methodology is very flexible, portable and economic, and probably the most*

*important aspect: very simple. As mentioned above, we wanted to put all the details*

*together in a single manuscript and we expect it to be valuable for the research*

*community.*

*Furthermore, as mentioned by the referee, our sampling method can collect large*

*biological material. While an upper cut size can be employed to increase the footprint*

*of the ecosystem represented, the approach used here is suitable for characterizing the*

*ecosystem of the immediate surrounding area which was our objective for this study.*

*As stated before, the aim of this article was not to provide answers on the chemistry*
*processes occurring in the atmosphere in a specific moment but to explain a method to*
*obtain the metabolomic profiles of the particles in suspension in the atmosphere. For*
*this reason, it is not necessary to sample a specific size range but that sample can*
*include all particles to obtain a general picture of which molecular compounds are*
*present in the particle fraction in the lower atmosphere. We are convinced that this*
*methodological approach provides very valuable information for the ecological*
*community.*

GC and LC results. LC results report 18 identified compounds. GC analysis mention 14
compounds. Most comprehensive aerosol analyses presented in the literature using these
techniques identify many more compounds. It is not clear why in the study presented here
only a small number of compounds was identified. There is no evidence given how these
compounds were identified. Simply mentioning "Library identification" is not sufficient. More
details would need to be given.

*The number of compounds identified and verified in a sample depends mainly on three*
*factors: i) the solvents used for the extraction of metabolites, ii) the concentration of*
*metabolites in the samples and iii) the specific metabolites present in the metabolite*
*databases.*

*Typically, un-targeted metabolomics techniques have been applied to obtain the*
*metabolic fingerprints and profiles from living organisms. For this reason the*
*metabolite databases include metabolites from living organisms and mainly from their*
*primary metabolism. The focus of our method are compounds known to be metabolites*
*coming from living organisms. According to the metabolite databases used for this*
*study we assigned a bit more than 30 compounds combining both GC and LC-MS*
*methodologies that are directly linked to the metabolism of organisms, likely from*
*plants. Additionally, sampling was performed in an area with very low biological*
*activity compared to more forested areas and organic volatile compounds derived from*
*plants or anthropogenic emissions could not be identified by our libraries since those*
*compounds are simply not normally listed in our metabolomic libraries. As we*
*mentioned in the manuscript: "The techniques to characterize the gas phase*
*component of atmo-metabolomes are well described elsewhere (Smith and Španěl,*

*2011; Tholl et al., 2006). Our purpose here is to describe an atmo-metabolomic method*

*for sampling aerosols and characterize the particle phase of the atmo-metabolomes.".*

*In the past version of the manuscript we also mentioned that "Metabolite assignation*

*with LC-MS was performed by our metabolite library with more than 200 typical*

*metabolites usually present in plants and fungi including products from primary and*

*secondary metabolism"; so we made it clear which kind of compounds we were*

*targeting. Our intention was never to reproduce a method to sample all atmospheric*

*organic compounds but to measure with LC-MS and GC-MS compounds present in solid*

*particles coming from living systems.*

*We have rewritten the section regarding the metabolite identification by LC-MS. It now*

*reads: "Metabolite assignation with LC-MS was performed by our metabolite library*

*with more than 200 typical metabolites usually present in plants and fungi including*

*products from primary and secondary metabolism. Assignation were performed*

*separately for each ionization mode (positive and negative) and using the exact mass*

*of metabolites and RT. ESI do not typically fragment all ions, however, in some*

*molecules we could still detect some fragments which were also considered for the*

*metabolite assignation and relative quantification. According to Sumner et al., 2007,*

*our LC-MS metabolite assignment is putative since it was based on total exact mass of*

*the metabolite and RT of standard measurements in the instrument. However, the use*

*of high MS resolution achieved with Orbitrap technology and RT reduces substantially*

*the number of false positive assignations. For more detailed information regarding the*

*metabolite assignation see Rivas-Ubach et al., (2016b). RT and m/z values of*

*metabolite matching for LC-MS are shown in Table S2."*

Figure 7 and 8 show interesting results but more discussion would be needed.

*Already responded above.*

Section 4.1 is mostly trivial discussion and can be shortened a lot. The same applies to much of section 4.3.

*Following the referee's advice, we have now shortened the section 4.1 and deleted the*

*section 4.3 re-organizing some content into the introduction.*

Anonymous Referee #2

Overview: This manuscript describes a metabolic-approach for the analysis of atmospheric
aerosol. The approach includes GC/MS, LC/MS and direct injection FT-ICRMS measurements.
To demonstrate the potential for this method to contribute toward an improved
understanding of natural metabolites associated with aerosol, the authors studied the
composition of aerosol collected in the spring and the summer. Key results include: the finding
that plant-related metabolites (namely organic acids and carbohydrates) are higher in the
spring than summer; the summer samples included metabolites associated with oxidative
stress; and summer aerosol composition included a higher fraction of high molecular weight
compounds than spring with a higher O/C ratio. The manuscript contains very valuable
laboratory method information that is well referenced. However, the details about the
advanced statistical analysis are deficient. The introduction and methods sections are well-
written, but the results and discussion section seems to be presented poorly. Given the
inadequate description of the statistical approach, I found the results section to be especially
difficult to understand. Another aspect for further consideration is placing this work into the
context of the current literature on aerosol chemistry. There's quite a bit of similar work
without a so-called "metabolomics" approach that is relevant.

*Many thanks for the positive evaluation on the interest of the study* and for considering
very valuable the laboratory method information. *We have now rewritten the*
*introduction, focusing it on the ecological applications of the study of the metabolomic*
*fingerprint of ecosystems on atmospheric aerosols. And to address the referee's*
*concerns, we have now clarified the statistical methods section and combined the*
*results and discussion. We hope that now the text is clearer.*

*We acknowledge that GC-MS and other mass spectrometry techniques have been*
*widely used in the atmospheric research. Nonetheless, as in our response to the*
*previous referee, our purpose for this manuscript was not to improve or replace those*
*well-defined approaches or to investigate the chemistry of the atmosphere. We present*
*an approach that is novel and useful for the ecological community by enabling*
*researchers to detect aerosol metabolites that may be directly linked with the main*
*physiological and ecological processes of living organisms.*

Specific suggestions:

The literature review of atmospheric aerosol composition is weak and outdated. Since the
authors claim to be the first to apply metabolomics techniques to aerosol, which are not
necessarily different from other composition measurements, it would be nice if they would
acknowledge the vast literature of GC/MS, LC/MS and FT-ICR-MS results aimed at
understanding aerosol composition.

*Our manuscript aims to describe in enough detail the set of necessary procedures to*
*obtain the metabolome profiles from aerosols. As mentioned above, this method is*
*mainly focused to detect signatures directly linked to the main physiological and*
*ecological processes of organisms; metabolites which are not volatile but are also part*
*of many particles in suspension in the atmosphere. We have modified the introduction*
*more clearly focusing the aims on the ecological aspects.*

Lines 102 - 106: How important is the carbon and nutrient deposition of aerosols to ecological
systems?

*We have expanded the section in the introduction about the aerosol deposition on*
*ecosystems.*

Lines 145 - 148: The atmospheric system is quite complex and the goals of this manuscript are
quite broad. I suggest some refinement of the manuscript goals with a focus on a well-defined
portion of the atmospheric system, since this work doesn't address larger spatial sampling,
research flight measurements, or multiphase measurements.

*As mentioned above, we have rewritten the introduction section. We have now better*
*focused our manuscript and hope the purposes of our method are clearer.*

Line 187: I often see this statement in manuscripts, but it is not a realistic resolving power for
environmental samples. Can the authors cite a paper demonstrating the successful
measurement of a complex mixture with a resolving power and actual resolution of 1,000,000?

*We reviewed the capability of FT-ICR-MS in the manuscript as an introduction of this*

*analytical method. Therefore, we have to report the maximum resolving power that FT-*

*ICR-MS can achieve. However, we did not state that such resolving power is currently*

*used in environmental study. We stated the actual resolving power (~400,000 at 400*

*m/z) for our samples.*

The organization of sections 2.3 - 2.5 is a little bit strange. Specifically, a description of the

GC/MS sample prep (in 2.3) is given followed by LC/MS analysis (2.4), which is in turn followed by the GC/MS analysis (2.5).

*We understand that this may create some confusion, however, we wanted to be*

*consistent and we have followed the same order for the methods and results along the*

*article; LC-MS, GC-MS and FTICR consecutively.*

*The section 2.3 described the extraction of metabolites from the quartz filters which is*

*common for all the three MS techniques (LC, GC and ICR). However, differently to LC-*

*MS and DI-FT-ICR extracts, samples for GC-MS require an additional step; the*

*derivatization of metabolites. This step is also indicated in the Figure 3 and it is clearly*

*linked to the extraction of metabolites. We considered the derivatization should not be*

*in the following section of GC-MS analyses (2.5). However, we could consider moving*

*this section if required.*

*After the section for sample preparation (2.3)(common for the three techniques), we*

*have described the parameters used for each one of the MS instruments separately*

*according to the order established (2.4 for LC-MS, 2.5 for GC-MS and 2.6 for DI-FTICR).*

*Following the instrument analysis sections, the next 3 sections (2.7, 2.8 and 2.9)*

*provide the details to obtain the numerical data from each of the instruments. Also,*

*these 3 sections follow the same order established, so 2.7 for LC-MS, 2.8 for GC-MS and*

*2.9 for FT-ICR.*

*So, our logic for the description of the methods was:*

*1. Extraction of metabolites. (2.3)*

*2. Data acquisition by each MS instrument (2.4, 2.5 and 2.6)*

*3. Processing of MS chromatograms/spectra from each instrument. (2.7, 2.8 and 2.9)*

*We think that this order is comprehensive; however, we can change the distribution of*
*the methods if required. An option would be to include all the instruments in a single*
*"Data acquisition" and "Processing of chromatograms/spectra" section by using*
*subtitles.*

Line 346: How were both positive and negative ionization performed with LC/MS? Were they
done in separate runs or using fast polarity switching?

*The LTQ Orbitrap Velos cannot switch ionization polarities quickly. Only the most recent*
*Q-Exactive and the new LUMOS Orbitrap versions can operate with fast polarity switch.*
*So samples were first injected in positive mode and then in negative mode. We now*
*have indicated this detail in the manuscript and it can be read as: "All samples were*
*first analyzed in positive (+) ionization mode and later in negative (-) ionization mode."*

Line 371: Was negative mode ESI performed? Why was negative ESI not performed for
atmospheric aerosol characterization?

*Analyses in FT-ICR-MS were performed exclusively in negative mode as already*
*mentioned in the manuscript: "Samples were directly infused into the mass*
*spectrometer using a standard Bruker electrospray ionization (ESI) in negative mode at*
*a flow rate of 3.0 μL/min through an Agilent 1200 series pump (Agilent Technologies,*
*Santa Clara, CA, USA ."*

*FT-ICR-MS in negative mode is the most used method to investigate natural organic*
*matter. While positive ESI mode could increase the compound coverage, we opted to*
*use negative mode only as our instrument was optimized under ESI(-) for organic*
*matter exploration.*

Lines 381 - 383: Both fragment ions and exact mass were used to assign metabolites. Were
these measurements made in single runs LTQ MS/MS and FT-MS in tandem or something else?

*Although we already referenced a manuscript where the metabolite assignation is well*
*described, we agree that this section should be more detailed, especially for a*
*methodological article like the present one. We have extended this section of the*

*manuscript. It now reads: "Metabolite assignation with LC-MS was performed by our*
*metabolite library with more than 200 typical metabolites usually present in plants and*
*fungi including products from primary and secondary metabolism. Assignation were*
*performed separately for each ionization mode (positive and negative) and using the*
*exact mass of metabolites and RT. ESI do not typically fragment all ions, however, in*
*some molecules we could still detect some fragments which were also considered for*
*the metabolite assignation and relative quantification. According to Sumner et al.,*
*2007, our LC-MS metabolite assignment is putative since it was based on total exact*
*mass of the metabolite and RT of standard measurements in the instrument. However,*
*the use of high MS resolution achieved with Orbitrap technology and RT reduces*
*substantially the number of false positive assignations. For more detailed information*
*regarding the metabolite assignation see Rivas-Ubach et al., (2016b). RT and m/z*
*values of metabolite matching for LC-MS are shown in Table S2."*

Line 418: Why was S/N > 7 used as a threshold? How was the S/N determined?

*S/N was determined in the Bruker Data Analysis software, which was assessed based*
*on baselines near each peak. S/N of 3 and 5 are often used in natural organic matter*
*exploration as that range is considered as the mínimum detection limit (Riedel and*
*Dittmar 2014). We chose S/N>7 for a more conservative measure.*

Lines 473 - 476: How many data points were used for this analysis? How were the sub-sets of
data selected for analysis? Some discussion on the QA filtering procedures and selection of
data for statistical analysis is greatly needed.

*Each analytical technique generated their own data that were posteriorly analyzed*
*separately. All the data (metabolomic fingerprints) from each instrument were used to*
*perform the PERMANOVAs. PERMANOVAs were performed separately.*

*As this manuscript was not especially focused on the understanding of the*
*metabolomes or chemical signatures between summer and spring aerosols, we did not*
*include the number of features we observed and used for the statistical analyses. We*
*can include this information if you think it necessary.*

*We have added some more text in the material and methods section explaining the*
*data filtering in more detail. Now the text reads: "For each season (spring and summer)*
*and dataset (LC-MS, GC-MS and FT-ICR-MS), the variables present in less than 50% of*
*the samples were excluded for the statistical analyses. The signal values measured in*
*the experimental blanks in each of the instruments were subtracted from the datasets.*
*Each of the variables from metabolome fingerprints obtained from each MS instrument*
*were posteriorly submitted to Levene's and Shapiro tests to assess homogeneity of*
*variances and normality, respectively. Variables that did not comply with those*
*statistical assumptions were removed from the datasets. Outlier measurements were*
*replaced for missing values and were defined as those measurements of a specific*
*variable with values three-fold higher than the third quartile or three-fold lower than*
*the first quartile of each season. For FT-ICR-MS datasets we have been very*
*conservative and only the formula assigned features that presented less than 0.3ppm*
*of error were used although cutoff values up to 0.5ppm are typically used (Osterholz et*
*al., 2016)."*

Line 487: In what sense is the statistical significance?

*As typically used in the vast majority of environmental studies, the alpha error or type I*
*error is maintained at 5%. The term "statistical significance" is widely used for P values*
*lower than 0.05 for a given test. So, alpha error (type I error), the probability of*
*rejecting the null hypothesis when is true, was maintained at 5%.*

Lines 489-496: What do these compounds indicate? How were they identified?

*In the results section we only indicate which compounds increased significantly (P*
*<0.05) or marginally significantly (P <0.1) in the spring samples. Some of the*
*metabolites identified are briefly discussed in the discussion section (4.4). We did not*
*discuss all the results obtained with each of the instruments since it would be out of the*
*main aim of the study. This article is just a methodological article and we have focused*
*the discussion on the major results and it was not our intention to investigate all of the*
*differences between the seasons.*

*Those compounds were identified according to our LC-MS database of metabolites of*

*plants and fungi, however, as already mentioned above we have now extended the*

*section of metabolite identification and provided more details.*

501-504: This approach from Kim et al. is highly speculative. It's also not an appropriate approach for atmospheric aerosol. Did you extra proteins? How did you verify protein-like components?

*We highly agree with the referee. Although the compound classification obtained from*

*van Krevelen (vK) diagrams (O:C vs. H:C) provides a certain approximation of the*

*composition of the samples, we also think that their use should be limited. However, vK*

*diagrams are widely used to understand the chemical changes in samples and this*

*classification is still widely used to represent the FTICR data. Because this compound*

*classification is a widely used method to understand organic matter composition, our*

*intention was to show this to the readers. However, it should be noted that even in the*

*previous manuscript version we only briefly mentioned this classification. In fact, we*

*are already working on another manuscript reviewing this commonly used compound*

*classification for FTICR data. For this reason, in the new version of the manuscript we*

*finally decided to retain the review of the existence of such classification but we have*

*deleted the previous Figure 7.*

Lines 517 - 520: What is the meaning of this observation?

*Here we mention that particles in summer showed significantly higher intensities in*

*features with higher O/C ratios. This result is briefly discussed in the discussion method,*

*however, as the aim of the article is solely methodological, we did not discuss each of*

*the results in depth. We simply chose two seasons to test if we could detect statistically*

*significant differences between the "atmo-metabolomes" between the two seasons.*

*Different factors could be chosen for this test, like two different ecosystems but we*

*considered that seasonality was more a feasible and comprehensive factor to test.*

Aerosol sampling information is vague and seems to imply that the authors are unfamiliar with
standard sampling techniques for atmospheric chemistry. How did you assess the total carbon
concentrations, filter artifacts, and other recovery issues?

*Our intention was not to reproduce a standard atmospheric chemistry sampling*
*technique as we recognize that there are numerous researchers focused on the*
*chemistry transformations in the atmosphere and for that reason many specific*
*protocols are typically used. However, our simple method is suitable for characterizing*
*the metabolome of the atmosphere.*

*The aim of metabolomics is to compare relatively different groups of samples. Since it*
*is practically impossible to obtain a full metabolome in terms of absolute*
*concentrations for each of the detected metabolites, as long as the sample preparation*
*is performed equally for all the samples we can perform a relative comparison between*
*groups of samples. Filter artifacts were coped with experimental blanks that were*
*injected to all instruments and any signal obtained from those blanks was posteriorly*
*subtracted from the original samples. The use of blanks is a standard procedure for*
*any metabolomics study. We have now included more details in the material and*
*methods to respond to those concerns.*

Sampling flow rates are expected to change with diurnal cycles (e.g., temperature & pressure);
how was this recorded or accounted for?

*Each filter was sampled exactly for the same amount of time and in the same time*
*range as described in the material and methods section: "The pump was working daily*
*during 18 consecutive hours and pumped air at 30 L per minute through each filter.*
*Filters were replaced manually before 09:00am and the pump started working*
*automatically at 09:00am and stopped automatically at 03:00am the following day.*
*Filters were stored at -80°C until metabolite extraction. Filters were sampled on a*
*tower at 8 meters height."*

Lines 535 - 537: The purpose of the study was to assess the sensitivity of different mass
spectrometry instruments. But, I didn't understand how that was accomplished? Did you define method detection limits or find any limitations in your approach? More discussion on this would be appreciated.

*We rely on the statistical analyses to test the sensitivity of the used techniques to*

*detect changes between seasons. We sampled in an area with a very low primary*

*producer activity and still we were able to detect significant differences in the overall*

*atmo-metabolomes between spring and summer. The significance obtained in the*

*PERMANOVA test proves that each of the techniques was sensitive enough to detect*

*changes between those samples. The principal component analyses (PCAs) for each of*

*the instruments also prove that the instruments were able to detect significantly*

*different overall composition in the spring vs. summer samples. In order to clarify this*

*concern, we have modified the text properly in different sections.*

How does you approach differ from the existing approaches to canopy measurements or other ecological studies focused on atmospheric-biosphere exchange?

*In this article we explained, and put together, the different steps to obtain the*

*metabolomic fingerpints (or metabolomic signatures) from particles sampled in the*

*lower atmosphere. As far as we know, no other approach for analyzing aerosol*

*metabolomes has been published.*

*Similar sampling methods can be performed in other ways with different pumps and*

*filters, however, the method we propose is more portable (lower weight and volume),*

*flexible (can be easily manipulated in different ways) and more economic than the*

*commonly commercialized prototypes for aerosol filter sampling. Also as a*

*methodological article we provided detailed information on how our sampling was*

*designed and performed. As discussed in the manuscript, the main idea is to obtain the*

*minimum values in the filter-size/pump-flow ratio to concentrate as much as possible*

*the filters. Our objective was not to perform a comparative study with all the available*

*sampling methods. We just described a very simple and flexible method that samples*

*particles in suspension efficiently and at a low cost. In addition, researchers can choose*

*the filter size they require while many commercial systems are compatible only with a*

*unique filter size.*

Lines 584-587: Which solvents did you use to sequentially extract the filters? How did you evaluate the results of various solvent combinations?

*We did not perform a sequential extraction in this study. We used methanol:water*

*(80:20) as one of the most widely used solvent mix for extraction of metabolites. We*

*cite different studies and methods where the number of extractions and recovery is*

*discussed. We did not attempt to use a whole variety of extraction methods; we only*

*aimed to show a generally used extraction method to investigate whether the*

*analytical techniques can differenciate statistically the metabolomes between spring*

*and summer aerosols. We also mention that this extraction method is not exclusive but*

*suggested and indicate that different extraction methods can be also used. As widely*

*discussed in several analytical chemistry articles, different extraction methods obtain*

*different range of metabolites based mainly on their polarity.*

Lines 590-591: What was quantified in your study?

*In this study we performed a relative quantification of the metabolomic fingerprints for*

*comparative analyses between spring and summer.*

Lines 596 - 600: How was the absorption extract recovery assessed?

*We measured how much volume of solvent was recovered (after the extraction*

*procedures) with respect to the initial solvent added. In the text we mention that we*

*can get an extraction recovery of 89% which indicates that we recover 0.89mL per each*

*1mL added to the tubes with the filters to perform the extraction. We did not think that*

*it was necessary to incorporate this information in the methods section. However, we*

*can introduce the explanation if the referee thinks it is necessary.*

Line 623: "match" or assign?

*We appreciate you made us notice this, we agree "assign" is more suitable than*

*"match" in this sentence. We have changed the word in the manuscript.*

Lines 706-710: Please clarify how the "metabolic fingerprint" was defined/classified?

*We have added a clarification of metabolomic fingerprints and metabolomic profiles in*
*the introduction section. Now the text reads: "The first step to characterize a*
*metabolome profile is to obtain the chemical signature of the sample (metabolomic*
*fingerprint) without further molecular identification (Sardans et al., 2011). The*
*identification of specific metabolites can be further obtained by the information*
*present in the metabolomic fingerprints. In this study, we describe the different*
*procedures to obtain the metabolomic fingerprints and identify molecular compounds*
*from aerosols. This atmo-ecometabolomics methodology is a potential tool to shed*
*light in novel questions in ecology, especially for the ecosystem-atmosphere interface."*

Table 1: Fingerprint information is unclear. Please add some explanation in the body of the
paper.

*See comment above.*

Figure 1: What about aqueous phase processing of VOCs or aerosol?

*We modified the figure but the different atmospheric VOCs transformations are not*
*presented in detail since it was never our intention to address that issue in this*
*manuscript.*

Figure 3: How were common inorganic ions removed from the samples before DI-FT- ICR-MS?

*It should be noted that most of the inorganic ions are at much lower mass range than*
*our FTICR-MS analytical window (100-1200 m/z). Thus, unless those ions generate*
*clusters that would interfere with the FT-ICR-MS measurements, such as sodium and*
*chloride, removal of inorganic ions were not necessary. In addition, such a problem is*
*more evident in direct infusion positive ion mode, which was not considered in this*
*study.*

Figure 5: I assume this is the list of "metabolic fingerprint" species. Please clarify.

*As mentioned before we have included the definition of what a metabolic fingerprint is.*

*The list of metabolites does not represent the entire fingerprint of the different seasons*

*but only the portion that has been identified/assigned. We hope it is now clearer.*

Figure 7: How were the species in (a) subsetted from the whole dataset?

*As explained before, we did not use a subset of the datasets but the whole amount of*

*detected features. However, we have now deleted this figure from the new manuscript*

*version.*

**Atmo-ecometabolomics: a new measurement approach for further investigate the link of atmospheric particles composition with ecosystem functioning.**

Albert Rivas-Ubach[1], Yina Liu[1], Jordi Sardans[2,3], Gourihar Kulkarni[4], Malak M. Tfaily[1], Young-Mo Kim[5], Eric Bourrianne[6], Ljiljana Paša-Tolić[1], Josep Peñuelas[2,3], Alex Guenther[7].

1. Environmental Molecular Sciences Laboratory, Pacific Northwest National Laboratory, Richland, 99354, WA, USA.
2. CREAF, Cerdanyola del Vallès, 08913 Catalonia, Spain
3. CSIC, Global Ecology Unit CREAF-CEAB-CSIC-UAB, Cerdanyola del Vallès, 08913 Catalonia, Spain
4. Atmospheric Sciences and Global Change Division, Pacific Northwest National Laboratory, Richland, WA, 99352, USA
5. Biological Sciences Division, Pacific Northwest National Laboratory, Richland, 99354, WA, USA.
6. Faculté des Sicences et d'Ingénierie, Université de Toulouse III Paul Sabatier, Toulouse, 31400, France.
7. Department of Earth System Science, University of California, Irvine, CA, USA

**Author of correspondence:**
Albert Rivas-Ubach
Environmental Molecular Sciences Laboratory,
Pacific Northwest National Laboratory,
Richland, WA, USA, 99354
Tel: 971 319 5962
e-mail: albert.rivas.ubach@pnnl.gov // albert.rivas.ubach@gmail.com

**Abstract.**

Aerosols directly and indirectly play crucial roles in the processes controlling the composition of the atmosphere and the functioning of ecosystems. Gaining a deeper understanding of the chemical composition of aerosols is beginning to be recognized as important for ecological research.  A comprehension of the chemical composition of aerosol particles chemistry can potentially provide valuable information to further understand the link between aerosols and ecosystems. In this study, we used mass spectrometry (MS) coupled to liquid chromatography (LC-MS), gas chromatography (GC-MS) and Fourier transform ion cyclotron resonance (FT-ICR-

MS) to describe step by step an efficient method to characterize the chemical composition of aerosols , namely the atmo-metabolome, from two distinct seasons: spring and summer. We used the data to test statistically whether the analytical platforms were sensitive enough as to detect overall differences between season atmo-metabolomes. Our results showed that our sampling and extraction methods are suitable for aerosol chemical characterization with any of the analytical platforms used in this study. The three datasets obtained from these individual platforms showed significant differences of the overall atmo-metabolome between spring and summer. LC-MS and GC-MS analyses identified several metabolites that can be attributed to pollen and other plant-related aerosols. Spring samples exhibit higher concentrations of metabolites linked to higher plant activity while summer samples had higher concentrations of metabolites that may reflect certain oxidative stresses. FT-ICR-MS analysis showed that summer aerosols were generally higher in molecular weight and with higher O/C ratios, indicating higher oxidation levels and condensation of compounds relative to spring. Our method represents advanced novel approach to study the link between the composition of aerosols and ecosystems.

**1. Introduction**

**1.1 Atmo-ecometabolomics.**

Aerosols are solids and/or liquids in suspension typically derived from both biogenic and anthropogenic sources (Canagaratna et al., 2007). Primary biological aerosol particles (PBAP) are directly released from organisms and include cells such as pollen, spores, or whole microorganisms as well as fragments from plants and animal debris (Després et al., 2012). Primary producers also produce large amounts of volatile organic compounds (VOCs) which are emitted into the atmosphere and together with anthropogenic sources, such as the combustion of fossil fuels, are oxidized and then condense forming secondary organic aerosols (SOA) (Després et al., 2012; Fuzzi et al., 2006; Kirkby et al., 2016; Pandis et al., 1992) (Figure 1). To date, most research have focused on how aerosols affect climate system and atmospheric processes (Andreae and Crutzen, 1997; Ayers and Gras, 1991; Baustian et al., 2012; Carlton et al., 2010; Després et al., 2012; Jokinen et al., 2015; Ramanathan et al., 2001; Zhang et al., 2004). However, the components of the biosphere, such as plants, are in constant interaction with aerosols and can play important roles in aquatic and terrestrial ecosystems at different levels (Baker et al., 2003; Gu et al., 2002; Mahowald et al., 2005; Seco et al., 2007). For example, aerosols can serve as important carbon and nutrient sources for the phyllosphere, which is the microbial communities coexisting in plant leaves (Arnold et al., 2000; Lindow and Brandl, 2003; Vorholt, 2012). The microbial diversity of the phyllosphere can produce a variety of effects on their hosts and therefore can affect the ecosystems (Peñuelas and Terradas, 2014; Whipps et al., 2008). Plants also can absorb deposited particles from the atmosphere (Fageria et al., 2009; Seco et al., 2007; Uzu et al., 2010; Wedding et al., 1975) but the effects of plant particle uptake has been mainly focused for for trace metals (Achotegui-Castells et al., 2013; Feng et al., 2011; Uzu et al., 2010; Xiong et al., 2014) and other significant nutrients for agricultural purposes (Fernández and Brown, 2013). In aquatic ecosystems, much research has focused on the aerosol deposition as the nutrient source for phytoplantkon (Baker et al., 2003; Paerl, 1997; Paytan et al., 2009; Wang et al., 2015). Aerosol deposition represents thus an important source of nutrients for ecosystems (Baker et al., 2003; Wang et al., 2015). It is has been widely studied that the different nutrient proportions, mainly C, N and P, can determine the ecosystem structure and function (Elser et al., 1996; Sterner and Elser, 2002) and any significant change in the composition of aerosols may produce significant to produce significant shits in ecosystems (Carnicer et al., 2015; Peñuelas et al., 2012; Sardans et al., 2012a).

Studies on VOCs have already addressed several atmosphere-ecosystem interface questions at chemical level (Kantsa et al., 2015; Seco et al., 2007, 2015). However, other low molecular weight metabolites (~80-1000 Da), directly derived from diverse primary and secondary physiological processes from living organisms, are not commonly identified or taken into account in aerosol particles and may play important roles in the ecosystem functioning.

Metabolomics aims to study the metabolome of entire organisms or specific cells or tissues and includes the all the used procedures for sample collection, metabolite extraction, extract analysis and data analysis (Figure 2). A metabolome consists of the thousands of small (< 1,000

Da) compounds (metabolites) present in an organism at a given time (Fiehn, 2002). Such molecules include the substrates and products of cellular primary metabolism such as sugars, amino acids, and nucleotides as well as the plant and fungi secondary metabolism compounds such as polyphenolics. Those metabolites are all involved in a great variety of complex physiological processes to maintain the organisms' homeostasis, growth and responses to biotic and non-biotic stressors (Peñuelas and Sardans, 2009). Metabolomic techniques have been widely applied in biomedicine (Claudino et al., 2007; Walsh et al., 2006), human nutrition (Gibney et al., 2005; Wishart, 2008), plant physiology (Hirai et al., 2004; Kaplan et al., 2004), and more recently in ecology (ecometabolomics)(Bundy et al., 2008; Rivas-Ubach et al., 2012;

Sardans et al., 2011) to understand how flexible are the metabolomes change under certain circumstances or stressors situations (Gargallo-Garriga et al., 2014; Rivas-Ubach et al., 2016a,

2016c, Sardans et al., 2011, 2014). The first step to characterize a metabolome profile is to obtain the chemical signature of the sample (metabolomic fingerprint) without further molecular identification (Sardans et al., 2011). The identification of specific metabolites can be further obtained by the information present in the metabolomic fingerprints. In this study, we describe the different procedures to obtain the metabolomic fingerprints and identify molecular compounds from aerosols: atmo-ecometabolomics. This methodology is a potential tool to shed light in novel questions in ecology, especially for the ecosystem-atmosphere interface.

In this study, we propose atmo-ecometabolomics as a novel tool to detect molecular signatures directly related to stress (biomarkes) at a very large environmental scale. Recent climate projections predict an enhancement of extreme climatic events such as warming and drought which will lead to increases in plant stress and BVOC emissions (Peñuelas and Staudt,

2010). Plants have shown large chemical composition shifts when exposed to environmental stressors (Leiss et al., 2009; Macedo, 2012; Rivas-Ubach et al., 2014, 2016b; Sardans et al.,

2011). Several stress biomarkers have been already identified (Glauser et al., 2008; Guy et al.,

2008; Henry J. Thompson et al., 2005; Keltjens and van Beusichem, 1998; Shulaev et al., 2008)

and could also be reflected in aerosols as indirect indicator of the stress status of ecosystems.

Moreover, significant shifts in phenology in ecosystems have been detected during the last decades (Menzel et al., 2006; Parmesan, 2006; Parmesan and Yohe, 2003; Walther et al., 2002).

According to the relationship between the phenological stage of ecosystems and the metabolic signatures in aerosols; long temporal atmo-ecometabolomics studies can also potentially provide crucial information of the phenological changes of ecosystems. Moreover, each ecosystem should present specific metabolomic signatures in aerosols which long temporal atmo-ecometabolomics studies could also provide important information of the succession or recession of ecosystems. Additionally, the large variety of compounds forming part to aerosol particles could be also of great interest because of their importance for human health including lung diseases and allergies (D'Amato et al., 2002; Després et al., 2012; Pope et al.,

1995) (Figure 1). Therefore, atmo-ecometabolomics may serve as a powerful tool to assess stress and phenological changes at ecosystem and larger scales through the characterization and quantification of metabolomic signatures and specific biomarkers. The aim of this study is to layout detailed procedures to define the metabolome from particles in suspension in the low atmosphere. Ecologists can thus benefit from this approach for investigating further the link between aerosol composition and ecosystems.

**1.2 Atmo-ecometabolomic analytical instruments.**

Mass spectrometry (MS) coupled to liquid or gas chromatographs (LC-MS and GC-MS

respectively) are recently the most common instruments for metabolomic analyses (Fiehn,

2002; Sardans et al., 2011; Zhang et al., 2012) demonstrating high performance and sensitivity (Pan and Raftery, 2007). LC-MS and GC-MS techniques provide a similar data format (or dimension) in metabolomic studies; i.e. in both techniques, metabolites are first separated through chromatography (liquid or gas) resulting in two independent and orthogonal values; mass-to-charge ratio (m/z) and retention time (RT) relative to each of the ions detected which are used to further improve the metabolite assignation (Sumner et al., 2007). Generally, in metabolomic studies, GC-MS is suitable for detecting compounds such as carbohydrates, fatty acids, essential oils, carotenoids and also organic acids (Gullberg et al., 2004). GC analyses present excellent reproducibility with minimal RT shifts between samples; however, GC-MS

requires sample derivatization which increases the labor time in sample preparation and provides indirect detection of the metabolites that complicates the elucidation of novel metabolites. LC-MS can cover plant secondary metabolites such as flavonoids, alkaloids, phenolic acids, and saponins together with primary metabolites such as amino acids, carbohydrates and organic acids (De Vos et al., 2007). LC techniques often show greater RT

shifts between samples but provides a direct detection of the metabolites since derivatization is not required. Nonetheless, no single mass spectrometry technique can cover all metabolite classes (Ding et al., 2007; Zhang et al., 2012), and the combination of platforms is a common approach in metabolomics to increase the number of metabolites measured in the metabolomes (Hall, 2006).

Mass resolving power of the spectrometers is an important factor to consider in metabolomics. The high-resolution of Orbitrap mass spectrometers reduces the error of metabolite matching considerably when using high-resolution metabolite libraries (Rivas-

Ubach et al., 2016b). FT-ICR-MS affords the highest mass resolving power (up to 1,000,000)

and thus enabling formula assignment of a wide range of detected compounds (Marshall et al.,

1998). Although FT-ICR-MS can be coupled to liquid chromatography, direct infusion ESI (DI) is the most common method to analyze samples with this technique. DI- FT-ICR-MS provides ultrahigh mass resolution (< 1 ppm mass error after internal calibration) that enables accurate elemental formula assignments to most of the detected compounds based on their exact mass alone (Klein et al., 2006; Kujawinski, 2002). As such, FT-ICR-MS provides powerful means to understand the global characteristics of any complex organic samples (Kim et al., 2003;

Reemtsma, 2009; Roullier-Gall et al., 2014; Schmitt-Kopplin et al., 2012; Sleighter and Hatcher,

2007; Tfaily et al., 2015). It should be noted that exact mass provided DI-FT-ICR-MS alone is not sufficient for putative metabolite identification, and peak intensity measured with such a method is only semi-quantitative (Kujawinski, 2002; Liu et al., 2015). However, it is possible to assess the diversity of molecular species with different essential nutrients such as nitrogen, phosphorus or sulfur. This is especially interesting to understand how the elemental assignation in aerosols shifts in response to environmental changes; an important issue for ecological stoichiometry studies (Rivas-Ubach et al., 2012; Sardans et al., 2012b; Sterner and

Elser, 2002). DI-FT-ICR-MS acquisition time is significantly shorter (typically between 5-15

minutes) than MS coupled to a LC or GC which it can take over 40 minutes per sample.

Visualization of FT-ICR data using van Krevelen diagrams (vK) based on O:C and H:C

ratios of the assigned features have been used in numerous studies to understand chemical compositions of diverse complex organic matrices (Kim et al., 2003; van Krevelen, 1950;

Schmitt-Kopplin et al., 2012). vK diagrams provide important information of the main chemical reactions such as methylation, demethylation, hydrogenation, hydration, condensation, oxidation or reduction of the detected ions (Kim et al., 2003). Additionally, plotting O/C vs. H/C

ratios of all of the assigned formulas can also provide an approximation of the compound classes present in the samples (Kim et al., 2003; Minor et al., 2014; Sleighter and Hatcher,

2007). However, compounds in the environment can easily be transformed or degraded, and thus change their O:C and H:C ratios compared to their original form. Consequently, while this classification can still provide a general idea of the organic compound compositions in aerosols, any compound classification based on stoichiometric constraints should be used with caution.

C number versus mass (CvM) can also be used to represent FT-ICR-MS data and provides crucial information on oxidation processes or molecular weight shifts when comparing two or more systems (Reemtsma, 2009). Therefore, FT-ICR-MS is a useful tool to obtain high- resolution metabolomic profiles and to gain a better understanding of the aerosol sources as well as their chemical transformation in the atmosphere.

**1.3 Testing atmo-ecometabolomics.**

The present article aims to describe step by step a method for sampling and characterize the particle phase of the atmo-metabolomes by contrasting two distinct seasons: spring and summer. We designed a simple aerosol sampling method and collected total aerosol particles (without any size cutoff) in spring and summer of 2015 at the Pacific Northwest National

Laboratory campus (Richland, WA, USA). We used those samples to adapt the existing metabolomics protocols to extract the metabolites from aerosols in solvents to posteriorly analyze them with; i) LC-MS, ii) GC-MS and iii) DI-FT-ICR-MS. The generated data with each of the instruments was analyzed following some basic statistical approximations typical for ecometabolomics and chemical characterization studies. The aerosol sampling method, the metabolite extraction procedures and some major metabolomic differences between spring and summer aerosols are detailed and discussed. The techniques to characterize the gas phase component of atmo-metabolomes are well described elsewhere (Smith and Španěl, 2011;

Tholl et al., 2006). The application of atmo-ecometabolomics in natural ecosystems represents a new approach in ecology to shed light in the understanding of the link between metabolic composition of aerosols and ecosystems. This novel method in ecological sciences allows understanding deeply recent research issues related with ecosystem stress, phyllosphere, ecological stoichiometry, ecosystem phenology, global change, among others.

[revised manuscript text omitted]
 and RT. ESI do not typically fragment all ions, however, in some molecules we could still detect some fragments which were also considered for the metabolite assignation and relative quantification. According to Sumner et al., 2007, our LC-MS metabolite assignment is putative since it was based on total exact mass of the metabolite and RT of standard measurements in the instrument. However, the use of high MS resolution achieved with Orbitrap technology and RT reduces substantially the number of false positive assignations. For more detailed information regarding the metabolite assignation see Rivas-Ubach et al., (2016b). RT and m/z values of metabolite matching for LC-MS are shown in Table S2.

**2.8 Processing of GC-MS chromatograms.**

GC-MS data was processed with two different software; MZmine and Metabolite Detector. MZmine 2.17 (Pluskal et al., 2010) was specifically used to obtain the metabolomic fingerprints from the additional sampled filters to test the sonication time and be thus more consistent with the LC-MS data. Parameters to get the numerical datasets with MZmine are shown in Table S3.

Metabolite Detector 2.5 (Hiller et al., 2009) was used to process the GC-MS raw data files from the spring and summer. First, "Agilent .D" files were converted to netCDF format using Agilent Chemstation and posteriorly converted to "bin" files using Metabolite Detector. Chromatograms were deconvoluted, aligned and the metabolites were autoassigned before exporting the datasets in CSV format. Briefly, retention indices (RI) of detected metabolites were calculated based on the analysis of the FAMEs mixture, followed by their chromatographic alignment across all analyses after deconvolution. Metabolites were initially identified by matching experimental spectra to PNNL increased version of FiehnLib (Kind et al., 2009), containing spectra and validated retention indices for over 850 metabolites, with probability threshold of 0.8. NIST14 GC-MS library was also used to cross-validate identification of metabolites by matching fragmented spectra. All metabolite identifications were manually validated to reduce deconvolution errors during automated data-processing and to eliminate false identifications. Parameters used in Metabolite detector are shown in table S4. Metabolite matching information in GC-MS is shown in Table S5.

**2.9 Processing of DI-FT-ICR spectra.**

The mass spectrum for each sample was averaged over 144 individual scans and then internally calibrated using an organic matter homologous series separated by 14 Da (–CH2 groups). The mass measurement accuracy was typically within 1 ppm for singly charged ions across a broad $m/z$ range (100-1100 $m/z$). DataAnalysis software (BrukerDaltonik version 4.2) was used to convert raw spectra to a list of m/z values applying FTMS peak picker with signal to noise (S/N) of 7, which is above the minimum detection limit for FT-ICR-MS for NOM (Riedel and Dittmar, 2014) and absolute intensity threshold of 100. Chemical formulas, containing C, H, O, N, S, and P, were then assigned using an in-house built software following the Compound Identification Algorithm (CIA), described by Kujawinski and Behn (2006). Chemical formulas were assigned based on the following criteria: S/N >7, mass measurement error <1 ppm. All observed ions in the spectra were singly charged as confirmed by the 1.0034 Da spacing found between isotopic forms of the same molecule (i.e., between $^{12}C_n$ and $^{12}C_{n-1}$–$^{13}C_1$).

**2.10 Statistical analyses.**

For each season (spring and summer) and dataset (LC-MS, GC-MS and FT-ICR-MS), the variables present in less than 50% of the samples were excluded for the statistical analyses.

The signal values measured in the experimental blanks in each of the instruments were subtracted from the datasets. Each of the variables from metabolome fingerprints obtained from each MS instrument were posteriorly submitted to Levene's and Shapiro tests to assess homogeneity of variances and normality, respectively. Variables that did not comply with those statistical assumptions were removed from the datasets. Outlier measurements were replaced for missing values and were defined as those measurements of a specific variable with values three-fold higher than the third quartile or three-fold lower than the first quartile of each season. For FT-ICR-MS datasets we have been very conservative and only the formula assigned features that presented less than 0.3ppm of error were used although cutoff values up to 0.5ppm are typically used (Osterholz et al., 2016).

The metabolome fingerprints from aerosols obtained from each instrument (3

independent datasets; LC-MS, GC-MS and DI-FT-ICR-MS) were tested by PERMANOVAs using the Bray Curtis distance to test for overall metabolomic differences between spring and summer (Table 1). The permutations were set at 10,000. Posteriorly, each metabolome fingerprint was also subjected to principal component analysis (PCA) to show in two dimensions the natural variability among the samples (van den Berg et al., 2006; Kim et al.,

2010) (Figure 4).

Heat-map plots for the assigned variables with LC-MS and GC-MS were plotted to show the relative concentration change of specific metabolites between spring and summer (Figure 5).

Each assigned variable was also submitted to t-student test with season as the categorical factor to test for statistical significance (Table S6).

We counted the proportions of formula classes from the FT-ICR-MS dataset (CHO, CHNO,

CHOS, CHNOS, CHNOSP, CHOSP, CHOP, CHNOP, CHNOPS and CHOPS) for each sample. All calculated proportions were transformed using *arcsin(rootsquare)* before submitting them separately to t-student tests with season (spring and summer) as the categorical factor to assess for statistical significance (Figure 6). A t-test was also performed on the O/C ratios of detected features in the FT-ICR-MS with season as the categorical factor to determine whether the oxidation status of the molecular compounds statistically change significantly between spring and summer (Figure 7).

The PERMANOVAs, PCAs, heat maps and t-tests were performed with R (R Core Team,

2013). The PERMANOVA analysis was conducted with the *adonis* function in the package

"vegan" (Oksanen et al., 2013). The PCAs were performed by the *pca* function of the

"mixOmics" package of R (Dejean et al., 2013). Heat maps were performed by the *heatmap.2*

function of the "gplots" package (Warnes et al., 2016). T-tests were performed with the function *t.test* in the package "stats" (R Core Team, 2013). All graphs were obtained by R and graphically treated by Adobe Illustrator CS6.

The value obtained from the deconvoluted peaks in LC-MS and GC-MS are directly related to the concentration of the corresponding variable even though they do not represent the real concentration in the sample in terms of mg of metabolite per weight of sample. However, the use of those values are suitable for metabolomic comparative analyses as previously demonstrated in other studies (Gargallo-Garriga et al., 2014; Lee and Fiehn, 2013; Leiss et al.,

2013; Mari et al., 2013; Rivas-Ubach et al., 2014, 2016c). In this study, we use the term *relative*

*abundance* when referring to the relative concentration of metabolites.

FT-ICR data is typically not directly quantifiable (Wozniak et al., 2008), however although not as robust than LC-MS or GC-MS techniques, using the intensity of the detected ions by FT-

ICR is still a good proxy of their relative concentration (Kellerman et al., 2014; Spencer et al.,

2015). We used the measured ion intensity for the specific vK and CvM representations, for those purposes the measured intensity of each individual ion detected in each of the samples was divided by the total intensity of the spectra (Kellerman et al., 2014; Spencer et al., 2015).

Chromatograms and spectra from LC-MS and FT-ICR-MS, respectively, of samples corresponding to days 16[th] and 30[th] June showed signs of contamination and were thus not considered in the corresponding datasets for statistical analyses.

**3. Results and discussion.**

**3.1 Aerosol sampling in filters and study site.**

Optimal flow rates for the aerosol collection is important; excessive flow rates may collapse the filters and low flow rates will not collect enough particles for a good detection of compounds. We used 37mm quartz filters that performed well without collapsing at flow rates of 50 L/min, however, after the internal tubing friction associated with the extension of the tubing and the sampling of two simultaneous filters caused a decrease in in the flow rates at the aerosol collection point and we achieved flow rates of 30 L/min. Larger tube diameters (>0.65cm diameter) could be considered when higher flow rates are necessary.

Our sampling method allows sampling different number of biological replicates at the same time for statistical purposes. Furthermore, sampling can be performed at different heights on a tower or mast by extending tubing if the pump performance is able to keep enough flow rates at the sampling point. So, the experimental design (number of replicates, filter material, length and diameter of tubing) and the pump performance are key elements to
consider in atmo-ecometabolomics.

Our aerosol collection was performed in a semi-urban area surrounded by landscapes
dominated by large and diverse agricultural cropland and a large desert shrubland with low
biological activity, so we expected to detect a complex variety of molecules that complicate
finding the atmospheric/ecological interpretation of the data. However, the obtained results
were equally useful to describe the main steps to obtain the atmo-metabolomes and to test
the sensitivity of different mass spectrometry techniques (LC-MS, GC-MS, FT-ICR-MS) to
characterize the atmo-metabolomes in low activity ecosystems and assess their potential for
detecting overall statistically significant changes between seasons.

**3.2 Metabolite extraction in organic solvents.**
Organic solvents combined with water are typically used for metabolomics analyses allowing
the extraction of a good range of semi-polar and non-polar metabolites (Kim et al., 2010; Lin et
al., 2006; Rivas-Ubach et al., 2013; t'Kindt et al., 2008). Solvents such as methanol, acetonitrile
or chloroform interact with plastics, especially under sonication, and chromatograms may
show contaminant features when using plastic tubes for metabolite extraction (Figure S2). Our
results showed that the use of silanized glass tubes is highly recommended during the
sonication step (Figure 3.4) to avoid artifacts. Combusted glassware for 5 hours at 450ºC or
higher is also recommended to prevent from any organic contaminants. If plastic tubes are
finally used during the extraction, especially during sonication, an initial test to detect any
potential plastic contaminant is recommended.

Methanol/water (80:20) solution typically used in metabolomics studies showing a
wide recovery of polar and semi-polar metabolites compared to other organic solvents (t'Kindt
et al., 2008), however, the use of other solvents recover different matrices of compounds. We
performed two extractions with the same solvent on the same sample to ensure higher
metabolite recovery from the aerosol samples (Böttcher et al., 2007; Nikiforova et al., 2005;
Rivas-Ubach et al., 2013, 2014) (Figure 3.5).

The filter size is also an important factor to consider for atmo-ecometabolomic
analyses. On one hand, the lower the ratio of *filter size/pump flow rate* is, the more
concentrated the samples will be. On the other hand, smaller filters are easier to handle in the
laboratory during extractions allowing also higher extract recovery. Quartz filters absorb high
volumes of extract that cannot be easily recovered. Our protocol with 37mm diameter filters
recovered the of 89% initial solvent volume. Larger filters complicate the extraction of metabolites (more filter handling, larger tubes and larger volumes of extract are required) and decrease considerably the recovery of extracts due the large solvent absorption.

**3.3 Testing atmo-ecometabolomics contrasting two distinct seasons.**

[revised manuscript text omitted]

Summer aerosols presented significantly higher proportions ($P < 0.05$) of CHO features than spring aerosols (Figure 6) and, in addition, we generally measured higher relative intensities in high-mass features in summer aerosols with respect to spring aerosols which presented higher relative intensities in lower-mass features (Figure 7a). In a CvM plot, at a given carbon number, the increase of nominal mass is contributed by heteroatoms (e.g. N, S, and O). We observed that summer had higher relative intensities of features with higher-mass than spring but with the same number of C (see region between dashed lines in Figure 7a). In addition, T-test on the O/C values of the formula-assigned features with season as categorical factor showed how summer had significantly higher relative intensities in features with higher

O/C ratios (more oxidized compounds) than spring (Figure 7b, c). This result is in accordance with the higher compound masses found in summer respect to spring for a same C-number (Figure 7a) suggesting that aerosol components in summer have higher oxidation rates. This trend could be related to higher levels of photochemical oxidants associated with warm sunny conditions and increased atmospheric photo-oxidation of aerosols (Obee and Hay, 1997).

Moreover, we also found higher relative intensities in high-mass aerosol compounds (over 500

Da) in summer (Figure 7a) which may suggest higher rates of polymerization or aerosol condensation. These observations point to one of the major challenges in utilizing atmo- ecometabolomic data which is the confounding effects of atmospheric processing of the original biogenic emissions.

**3.4 Conclusions and future perspectives.**

· Although the sampling was performed in a complex region with an urban area surrounded by a rural desert landscape with relatively low biological activity, all mass spectrometry techniques (LC-MS, GC-MS and DI-FT-ICR-MS) still detected significant differences between the spring and summer aerosol metabolomes though the methanol/water (80:20) extraction.

· There is no unique analytical technique able to characterize the whole metabolome fingerprint of aerosols. LC-MS and GC-MS and the use of metabolite libraries allow us to detect specific molecular compounds in aerosols while DI-FT-ICR-MS allows obtaining quickly a high- resolution metabolic fingerprint providing the elemental composition of aerosol compounds

· Coupling environmental variables with atmo-ecometabolomics would allow a more precise interpretation of the link between biological systems and the aerosol composition.

· Long term atmo-ecometabolomic experiments in natural ecosystems would improve understanding of the seasonal and interannual shifts of the composition of aerosols, directly linking atmospheric composition with plant phenology and physiology, along natural gradients or environmental changes.

· The use of atmo-ecometabolomic techniques ecological sciences could improve the detection, identification and quantification of any molecular compound related with environmental stressors (biomarkers) providing important information of the general status of the ecosystems. A good description of such biomarkers and other relevant metabolites would allow the creation of aerosol compound libraries which could be applied to understand the status of ecosystems and provide a relatively simple and quick environmental assessment and monitoring tool.

· The study of the impacts of aerosols on the phyllosphere and/or the stoichiometry of ecosystems could be significantly improved by the understanding of the composition of aerosols.

· New modern instruments such as GC-MS Orbitrap should be implemented in atmo- ecometabolomic studies to enable high performance for both RT and m/z resolution. Advances in methodologies for metabolomic analyses, such as Ion Mobility Spectrometry coupled to mass spectrometers (IMS-MS), could potentially improve significantly the number of detected metabolites in aerosols from the current tens and hundreds to thousands.

**Acknowledgements.**

The authors thank Therese Clauss and Rosalie Chu for their laboratory support. This research was performed using EMSL, a DOE Office of Science User Facility sponsored by the Office of

Biological and Environmental Research at Pacific Northwest National Laboratory and by the

European Research Council Synergy grant SyG-2013-610028 IMBALANCE-P, the Spanish

Government projects CGL2013-48074-P and the Catalan Government project SGR 2014-274.

**References.**

Achotegui-Castells, A., Sardans, J., Ribas, À. and Peñuelas, J.: Identifying the origin of atmospheric inputs of trace elements in the Prades Mountains (Catalonia) with bryophytes, lichens, and soil monitoring, Environ. Monit. Assess., 185(1), 615–629, 2013.

Andreae, M. 0 and Crutzen, P. J.: Atmospheric Aerosols: Biogeochemical Sources and Role in

Atmospheric Chemistry, Science (80-. )., 276(5315), 1052–1058, 1997.

Arnold, A. E., Maynard, Z., Gilbert, G. S., Coley, P. D. and Kursar, T. A.: Are tropical fungal endophytes hyperdiverse?, Ecol. Lett., 3(4), 267–274, 2000.

Ayers, G. P. and Gras, J. L.: Seasonal relationship between cloud condensation nuclei and aerosol methanesulphonate in marine air, Nature, 353(6347), 834–835, 1991.

Baker, A. R., Kelly, S. D., Biswas, K. F., Witt, M. and Jickells, T. D.: Atmospheric deposition of nutrients to the Atlantic Ocean, Geophys. Res. Lett., 30(24), 2003.

Baustian, K. J., Cziczo, D. J., Wise, M. E., Pratt, K. A., Kulkarni, G., Hallar, A. G. and Tolbert, M. A.:

Importance of aerosol composition, mixing state, and morphology for heterogeneous ice nucleation: A combined field and laboratory approach, J. Geophys. Res. Atmos., 117(D6), 2012.

van den Berg, R. A., Hoefsloot, H. C., Westerhuis, J. A., Smilde, A. K. and van der Werf, M. J.:

Centering, scaling, and transformations: improving the biological information content of metabolomics data, BMC Genomics, 7(1), 142, 2006.

Böttcher, C., Roepenack-Lahaye, E. v., Willscher, E., Scheel, D. and Clemens, S.: Evaluation of

Matrix Effects in Metabolite Profiling Based on Capillary Liquid Chromatography Electrospray

Ionization Quadrupole Time-of-Flight Mass Spectrometry, Anal. Chem., 79(4), 1507-1513, 2007.

Bundy, J. G., Davey, M. P. and Viant, M. R.: Environmental metabolomics: a critical review and future perspectives, Metabolomics, 5(1), 3–21, 2009.

Canagaratna, M. R., Jayne, J. T., Jimenez, J. L., Allan, J. D., Alfarra, M. R., Zhang, Q., Onasch, T. B., Drewnick, F., Coe, H., Middlebrook, A., Delia, A., Williams, L. R., Trimborn, A. M., Northway, M. J., DeCarlo, P. F., Kolb, C. E., Davidovits, P. and Worsnop, D. R.: Chemical and microphysical characterization of ambient aerosols with the aerodyne aerosol mass spectrometer, Mass Spectrom. Rev., 26(2), 185–222, 2007.

Carlton, A. G., Pinder, R. W., Bhave, P. V. and Pouliot, G. A.: To What Extent Can Biogenic SOA be Controlled?, Environ. Sci. Technol., 44(9), 3376–3380, 2010.

Carnicer, J., Sardans, J., Stefanescu, C., Ubach, A., Bartrons, M., Asensio, D. and Peñuelas, J.: Global biodiversity, stoichiometry and ecosystem function responses to human-induced C–N–P imbalances, J. Plant Physiol., 172, 82–91, 2015.

Claudino, W. M., Quattrone, A., Biganzoli, L., Pestrin, M., Bertini, I. and Di Leo, A.: Metabolomics: Available Results, Current Research Projects in Breast Cancer, and Future Applications, J. Clin. Oncol., 25(19), 2840–2846, 2007.

D'Amato, G., Liccardi, G., D'Amato, M. and Cazzola, M.: Outdoor air pollution, climatic changes and allergic bronchial asthma., Eur. Respir. J., 20(3), 763–76, 2002.

Dejean, S., Gonzalez, I. and Le Cao, K.: mixOmics: Omics Data Integration Project, 2013.

Després, V. R., Alex Huffman, J., Burrows, S. M., Hoose, C., Safatov, A. S., Buryak, G., Fröhlich-Nowoisky, J., Elbert, W., Andreae, M. O., Pöschl, U. and Jaenicke, R.: Primary biological aerosol particles in the atmosphere: a review, Tellus B, 64(15598), 1–58, 2012.

Ding, J., Sorensen, C. M., Zhang, Q., Jiang, H., Jaitly, N., Livesay, E. A., Shen, Y., Smith, R. D. and Metz, T. O.: Capillary LC Coupled with High-Mass Measurement Accuracy Mass Spectrometry for Metabolic Profiling, Anal. Chem., 79(16), 6081–6093, 2007.

Elser, J. J., Dobberfuhl, D. R., MacKay, N. A. and Schampel, J. H.: Organism Size, Life History, and N:P Stoichiometry, Bioscience, 46, 674–684, 1996.

Fageria, N. K., Filho, M. P. B., Moreira, A. and Guimarães, C. M.: Foliar Fertilization of Crop Plants, J. Plant Nutr., 32(6), 1044–1064, 2009.

Feng, J., Wang, Y., Zhao, J., Zhu, L., Bian, X. and Zhang, W.: Source attributions of heavy metals in rice plant along highway in Eastern China, J. Environ. Sci., 23(7), 1158–1164, 2011.

Fernández, V. and Brown, P. H.: From plant surface to plant metabolism: the uncertain fate of
foliar-applied nutrients, Front. Plant Sci., 4, 289, 2013.

Fiehn, O.: Metabolomics - the link between genotypes and phenotypes, Plant Mol. Biol., 48(1–
2), 155–171, 2002.

Fuzzi, S., Andreae, M. O., Huebert, B. J., Kulmala, M., Bond, T. C., Boy, M., Doherty, S. J.,
Guenther, A., Kanakidou, M., Kawamura, K., Kerminen, V.-M., Lohmann, U., Russell, L. M. and
Pöschl, U.: Critical assessment of the current state of scientific knowledge, terminology, and
research needs concerning the role of organic aerosols in the atmosphere, climate, and global
change, Atmos. Chem. Phys., 6(7), 2017–2038, 2006.

Gargallo-Garriga, A., Sardans, J., Pérez-Trujillo, M., Rivas-Ubach, A., Oravec, M., Vecerova, K.,
Urban, O., Jentsch, A., Kreyling, J. and Beierkuhnlein, C.: Opposite metabolic responses of
shoots and roots to drought, Sci. Rep., 4, 6829, 2014.

Ghasemzadeh, A. and Ghasemzadeh, N.: Flavonoids and phenolic acids: Role and biochemical
activity in plants and human, J. Med. Plants Res., 5(31), 6697–6703, 2011.

Gibney, M. J., Walsh, M., Brennan, L., Roche, H. M., German, B. and van Ommen, B.:
Metabolomics in human nutrition: opportunities and challenges., Am. J. Clin. Nutr., 82(3), 497–
503, 2005.

Glauser, G., Guillarme, D., Grata, E., Boccard, J., Thiocone, A., Carrupt, P.-A., Veuthey, J.-L.,
Rudaz, S. and Wolfender, J.-L.: Optimized liquid chromatography–mass spectrometry approach
for the isolation of minor stress biomarkers in plant extracts and their identification by
capillary nuclear magnetic resonance, J. Chromatogr. A, 1180(1), 90–98, 2008.

Gu, L., Baldocchi, D., Verma, S. B., Black, T. A., Vesala, T., Falge, E. M. and Dowty, P. R.:
Advantages of diffuse radiation for terrestrial ecosystem productivity, J. Geophys. Res. Atmos.,
107(D6), 4050, 2002.

Gullberg, J., Jonsson, P., Nordström, A., Sjöström, M. and Moritz, T.: Design of experiments: an
efficient strategy to identify factors influencing extraction and derivatization of Arabidopsis
thaliana samples in metabolomic studies with gas chromatography/mass spectrometry, Anal.
Biochem., 331(2), 283–295, 2004.

Guy, C., Kaplan, F., Kopka, J., Selbig, J. and Hincha, D. K.: Metabolomics of temperature stress,
Physiol. Plant., 132(2), 220-235, 2008.

Hall, R. D.: Plant metabolomics: from holistic hope, to hype, to hot topic, New Phytol., 169(3),

453–468, 2006.

Thompson, H. T, Heimendinger, J., Gillette, C., Sedlacek, S. M., Haegele, A., O'Neill, C., and

Wolfe, P.: In Vivo Investigation of Changes in Biomarkers of Oxidative Stress Induced by Plant

Food Rich Diets, J. Agric. Food Chem, 53(15), 6126-6132, 2005.

Hiller, K., Hangebrauk, J., Jäger, C., Spura, J., Schreiber, K. and Schomburg, D.:

MetaboliteDetector: Comprehensive Analysis Tool for Targeted and Nontargeted GC/MS Based

Metabolome Analysis, Anal. Chem., 81(9), 3429–3439, 2009.

Hirai, M. Y., Yano, M., Goodenowe, D. B., Kanaya, S., Kimura, T., Awazuhara, M., Arita, M.,

Fujiwara, T. and Saito, K.: From The Cover: Integration of transcriptomics and metabolomics for understanding of global responses to nutritional stresses in Arabidopsis thaliana, Proc. Natl.

Acad. Sci., 101(27), 10205–10210, 2004.

Ingram, J. and Bartels, D.: The molecular basis of dehydration tolerance in plants, Annu. Rev.

Plant Biol., 47(1), 377–403, 1996.

Jokinen, T., Berndt, T., Makkonen, R., Kerminen, V.-M., Junninen, H., Paasonen, P., Stratmann,

F., Herrmann, H., Guenther, A. B., Worsnop, D. R., Kulmala, M., Ehn, M. and Sipilä, M.:

Production of extremely low volatile organic compounds from biogenic emissions: Measured yields and atmospheric implications, Proc. Natl. Acad. Sci., 112(23), 7123–7128, 2015.

Kantsa, A., Sotiropoulou, S., Vaitis, M. and Petanidou, T.: Plant Volatilome in Greece: a Review on the Properties, Prospects, and Chemogeography, Chem. Biodivers., 12(10), 1466–1480,

2015.

Kaplan, F., Kopka, J., Haskell, D. W., Zhao, W., Schiller, K. C., Gatzke, N., Sung, D. Y. and Guy, C.

L.: Exploring the Temperature-Stress Metabolome of Arabidopsis, Plant Physiol., 136(4), 4159–

4168, 2004.

Kellerman, A. M., Dittmar, T., Kothawala, D. N. and Tranvik, L. J.: Chemodiversity of dissolved organic matter in lakes driven by climate and hydrology, Nat. Commun., 5, 3804, 2014.

Keltjens, W. G. and van Beusichem, M. L.: Phytochelatins as biomarkers for heavy metal stress in maize (Zea mays L.) and wheat (Triticum aestivum L.): combined effects of copper and cadmium, Plant Soil, 203(1), 119–126, 1998.

Kim, H. K., Choi, Y. H. and Verpoorte, R.: NMR-based metabolomic analysis of plants., Nat.

Protoc., 5(3), 536–49, 2010.

Kim, J. H., Lee, B. C., Kim, J. H., Sim, G. S., Lee, D. H., Lee, K. E., Yun, Y. P. and Pyo, H. B.: The isolation and antioxidative effects of vitexin from Acer palmatum, Arch. Pharm. Res., 28(2),

195–202, 2005.

Kim, S., Kramer, R. W. and Hatcher, P. G.: Graphical Method for Analysis of Ultrahigh-

Resolution Broadband Mass Spectra of Natural Organic Matter, the Van Krevelen Diagram,

Anal. Chem., 75(20), 5336–5344, 2003.

Kim, Y.-M., Nowack, S., Olsen, M. T., Becraft, E. D., Wood, J. M., Thiel, V., Klapper, I., Kühl, M.,

Fredrickson, J. K., Bryant, D. A., Ward, D. M. and Metz, T. O.: Diel metabolomics analysis of a hot spring chlorophototrophic microbial mat leads to new hypotheses of community member metabolisms., Front. Microbiol., 6, 209, 2015.

Kind, T., Wohlgemuth, G., Lee, D. Y., Lu, Y., Palazoglu, M., Shahbaz, S. and Fiehn, O.: FiehnLib:

Mass Spectral and Retention Index Libraries for Metabolomics Based on Quadrupole and Time- of-Flight Gas Chromatography/Mass Spectrometry, Anal. Chem., 81(24), 10038–10048, 2009.

Kirkby, J., Duplissy, J., Sengupta, K., Frege, C., Gordon, H., Williamson, C., Heinritzi, M., Simon,

M., Yan, C., Almeida, J., Tröstl, J., Nieminen, T., Ortega, I. K., Wagner, R., Adamov, A., Amorim,

A., Bernhammer, A.-K., Bianchi, F., Breitenlechner, M., Brilke, S., Chen, X., Craven, J., Dias, A.,

Ehrhart, S., Flagan, R. C., Franchin, A., Fuchs, C., Guida, R., Hakala, J., Hoyle, C. R., Jokinen, T.,

Junninen, H., Kangasluoma, J., Kim, J., Krapf, M., Kürten, A., Laaksonen, A., Lehtipalo, K.,

Makhmutov, V., Mathot, S., Molteni, U., Onnela, A., Peräkylä, O., Piel, F., Petäjä, T., Praplan, A.

P., Pringle, K., Rap, A., Richards, N. A. D., Riipinen, I., Rissanen, M. P., Rondo, L., Sarnela, N.,

Schobesberger, S., Scott, C. E., Seinfeld, J. H., Sipilä, M., Steiner, G., Stozhkov, Y., Stratmann, F.,

Tomé, A., Virtanen, A., Vogel, A. L., Wagner, A. C., Wagner, P. E., Weingartner, E., Wimmer, D.,

Winkler, P. M., Ye, P., Zhang, X., Hansel, A., Dommen, J., Donahue, N. M., Worsnop, D. R.,

Baltensperger, U., Kulmala, M., Carslaw, K. S. and Curtius, J.: Ion-induced nucleation of pure biogenic particles, Nature, 533(7604), 521–526, 2016.

Klein, G. C., Rodgers, R. P. and Marshall, A. G.: Identification of hydrotreatment-resistant heteroatomic species in a crude oil distillation cut by electrospray ionization FT-ICR mass spectrometry, Fuel, 85(14), 2071–2080, 2006.

van Krevelen, D.: Graphical-statistical method for the study of structure and reaction processes of coal, Fuel, 29, 269–284, 1950.

Kujawinski, E.: Electrospray Ionization Fourier Transform Ion Cyclotron Resonance Mass
Spectrometry (ESI FT-ICR MS): Characterization of Complex Environmental Mixtures, Environ.
Forensics, 3(3), 207–216, 2002.

Kujawinski, E. B. and Behn, M. D.: Automated Analysis of Electrospray Ionization Fourier
Transform Ion Cyclotron Resonance Mass Spectra of Natural Organic Matter, Anal. Chem.,
78(13), 4363–4373, 2006.

Lee, D. Y. and Fiehn, O.: Metabolomic response of Chlamydomonas reinhardtii to the inhibition
of target of rapamycin (TOR) by rapamycin., J. Microbiol. Biotechnol., 23(7), 923–31, 2013.

Leiss, K. A., Choi, Y. H., Abdel-Farid, I. B., Verpoorte, R. and Klinkhamer, P. G. L.: NMR
metabolomics of thrips (Frankliniella occidentalis) resistance in Senecio hybrids., J. Chem. Ecol.,
35(2), 219–29, 2009.

Leiss, K. A., Cristofori, G., van Steenis, R., Verpoorte, R. and Klinkhamer, P. G. L.: An eco-
metabolomic study of host plant resistance to Western flower thrips in cultivated, biofortified
and wild carrots., Phytochemistry, 93, 63–70, 2013.

Lin, C. Y., Viant, M. R. and Tjeerdema, R. S.: Metabolomics: Methodologies and applications in
the environmental sciences, J. Pestic. Sci., 31(3), 245–251, 2006.

Lindow, S. E. and Brandl, M. T.: Microbiology of the phyllosphere., Appl. Environ. Microbiol.,
69(4), 1875–83, 2003.

Liu, Y., Kujawinski, E. B., McNutt, M., Camilli, R., Crone, T., Guthrie, G., Hsieh, P., Ryerson, T.,
Peacock, E., Nelson, R., Solow, A., Warren, J., Baker, J., Reddy, C., Camilli, R., Reddy, C., Yoerger,
D., Mooy, B. Van, Jakuba, M., Kinsey, J., Reddy, C., Arey, J., Seewald, J., Sylva, S., Lemkau, K.,
Nelson, R., Ryerson, T., Camilli, R., Kessler, J., Kujawinski, E., Reddy, C., Valentine, D., Peterson,
C., Rice, S., Short, J., Esler, D., Bodkin, J., Ballachey, B., Aeppli, C., Carmichael, C., Nelson, R.,
Lemkau, K., Graham, W., Redmond, M., Atlas, R., Hazen, T., Incardona, J., Vines, C., Linbo, T.,
Myers, M., Sloan, C., Anulacion, B., Stanford, L., Kim, S., Klein, G., Smith, D., Rodgers, R.,
Marshall, A., Hughey, C., Rodgers, R., Marshall, A., Qian, K., Robbins, W., Marshall, A., Rodgers,
R., McKenna, A., Nelson, R., Reddy, C., Savory, J., Kaiser, N., Fitzsimmons, J., Koch, B., Witt, M.,
Engbrodt, R., Dittmar, T., Kattner, G., Kujawinski, E., Kujawinski, E., Behn, M., Sleighter, R.,
Hatcher, P., Reemtsma, T., Kujawinski, E., Longnecker, K., Blough, N., Vecchio, R., Finlay, L.,
Kitner, J., McKenna, A., Williams, J., Putman, J., Aeppli, C., Reddy, C., Valentine, D., Hughey, C.,
Rodgers, R., Marshall, A., Walters, C., Qian, K., et al.: Chemical Composition and Potential

Environmental Impacts of Water-Soluble Polar Crude Oil Components Inferred from ESI FT-ICR

MS, edited by A. Quigg, PLoS One, 10(9), e0136376, 2015.

Macedo, A. F.: Abiotic Stress Responses in Plants: Metabolism to Productivity, in Abiotic Stress

Responses in Plants, pp. 41–61, Springer New York, New York, NY., 2012.

Mahowald, N. M., Artaxo, P., Baker, A. R., Jickells, T. D., Okin, G. S., Randerson, J. T. and

Townsend, A. R.: Impacts of biomass burning emissions and land use change on Amazonian atmospheric phosphorus cycling and deposition, Global Biogeochem. Cycles, 19, GC4030, 2005.

Mari, A., Lyon, D., Fragner, L., Montoro, P., Piacente, S., Wienkoop, S., Egelhofer, V. and

Weckwerth, W.: Phytochemical composition of Potentilla anserina L. analyzed by an integrative GC-MS and LC-MS metabolomics platform., Metabolomics, 9(3), 599–607, 2013.

Marshall, A. G., Hendrickson, C. L. and Jackson, G. S.: FOURIER TRANSFORM ION CYCLOTRON

RESONANCE MASS SPECTROMETRY: A PRIMER, Mass Spectrom. Rev., 17, 1–35, 1998.

Menzel, A., Sparks, T. H., Estrella, N., Koch, E., Aasa, A., Ahas, R., Alm-Kübler, K., Bissolli, P.,

Braslavská, O., Briede, A., Chmielewski, F. M., Crepinsek, Z., Curnel, Y., Dahl, Å., Defila, C.,

Donnelly, A., Filella, Y., Jatczak, K., Måge, F., Mestre, A., Nordli, Ø., Peñuelas, J., Pirinen, P.,

Remišová, V., Scheifinger, H., Striz, M., Susnik, A., Van Vliet, A. J. H., Wielgolaski, F.-E., Zach, S.

and Zust, A.: European phenological response to climate change matches the warming pattern,

Glob. Chang. Biol., 12(10), 1969–1976, 2006.

Minor, E. C., Swenson, M. M., Mattson, B. M. and Oyler, A. R.: Structural characterization of dissolved organic matter: a review of current techniques for isolation and analysis, Environ. Sci.

Process. Impacts, 16(9), 2064–2079, 2014.

Nikiforova, V. J., Kopka, J., Tolstikov, V., Fiehn, O., Hopkins, L., Hawkesford, M. J., Hesse, H. and

Hoefgen, R.: Systems Rebalancing of Metabolism in Response to Sulfur Deprivation, as

Revealed by Metabolome Analysis of Arabidopsis Plants, Plant Physiol., 138(1), 304–318, 2005.

Obee, T. N. and Hay, S. O.: Effects of Moisture and Temperature on the Photooxidation of

Ethylene on Titania, Environ. Sci. Technol., 31(7), 2034–2038, 1997.

Oksanen, J., Guillaume-Blanchet, F., Kindt, R., Legendre, P., Minchin, P., O'Hara, R., Simpson, G.,

Solymos, P., Stevens, M. and Wagner, H.: vegan: Community Ecology Package., 2013.

Osterholz, H., Singer, G., Wemheuer, B., Daniel, R., Simon, M., Niggemann, J. and Dittmar, T.:

Deciphering associations between dissolved organic molecules and bacterial communities in a pelagic marine system, ISME J., 10, 1717-1730, 2016.

Paerl, H. W.: Coastal eutrophication and harmful algal blooms: Importance of atmospheric deposition and groundwater as "new" nitrogen and other nutrient sources, Limnol. Oceanogr., 42(5part2), 1154–1165, 1997.

Pan, Z. and Raftery, D.: Comparing and combining NMR spectroscopy and mass spectrometry in metabolomics, Anal. Bioanal. Chem., 387(2), 525–527, 2007.

Pandis, S. N., Harley, R. A., Cass, G. R. and Seinfeld, J. H.: Secondary organic aerosol formation and transport, Atmos. Environ. Part A. Gen. Top., 26(13), 2269–2282, 1992.

Parmesan, C.: Ecological and Evolutionary Responses to Recent Climate Change, Annu. Rev. Ecol. Evol. Syst., 37(1), 637–669, 2006.

Parmesan, C. and Yohe, G.: A globally coherent fingerprint of climate change impacts across natural systems, Nature, 421(6918), 37–42, 2003.

Paytan, A., Mackey, K. R. M., Chen, Y., Lima, I. D., Doney, S. C., Mahowald, N., Labiosa, R. and Post, A. F.: Toxicity of atmospheric aerosols on marine phytoplankton., Proc. Natl. Acad. Sci. U. S. A., 106(12), 4601–4605, 2009.

Peñuelas, J. and Sardans, J.: Ecological metabolomics, Chem. Ecol., 25(4), 305–309, 2009.

Peñuelas, J. and Staudt, M.: BVOCs and global change, Trends Plant Sci., 15(3), 133–144, 2010.

Peñuelas, J. and Terradas, J.: The foliar microbiome, Trends Plant Sci., 19(5), 278–280, 2014.

Peñuelas, J., Munné-Bosch, S., Llusià, J. and Filella, I.: Leaf reflectance and photo- and antioxidant protection in field-grown summer-stressed Phillyrea angustifolia. Optical signals of oxidative stress?, New Phytol., 162(1), 115–124, 2004.

Peñuelas, J., Sardans, J., Rivas-ubach, A. and Janssens, I. A.: The human-induced imbalance between C, N and P in Earth's life system, Glob. Chang. Biol., 18(1), 3–6, 2012.

[revised manuscript text omitted]

2499, 2007.

Seco, R., Karl, T., Guenther, A., Hosman, K. P., Pallardy, S. G., Gu, L., Geron, C., Harley, P. and

Kim, S.: Ecosystem-scale volatile organic compound fluxes during an extreme drought in a broadleaf temperate forest of the Missouri Ozarks (central USA), Glob. Chang. Biol., 21(10),

3657–3674, 2015.

Seigler, D. S.: Shikimic Acid Pathway, in Plant Secondary Metabolism, pp. 94–105, Springer US,

Boston, MA., 1998.

Shulaev, V., Cortes, D., Miller, G. and Mittler, R.: Metabolomics for plant stress response.,
Physiol. Plant., 132(2), 199–208, 2008.

Sleighter, R. L. and Hatcher, P. G.: The application of electrospray ionization coupled to
ultrahigh resolution mass spectrometry for the molecular characterization of natural organic
matter., J. Mass Spectrom., 42(5), 559–74, 2007.

Smith, D. and Španěl, P.: Direct, rapid quantitative analyses of BVOCs using SIFT-MS and PTR-
MS obviating sample collection, TrAC Trends Anal. Chem., 30(7), 945–959, 2011.

Solberg, Y. and Remedios, G.: Chemical composition of pure and Bee-Collected pollen, Meld.
fra Norges landbrukshogskole, 59, 1–13, 1980.

Spencer, R. G. M., Mann, P. J., Dittmar, T., Eglinton, T. I., McIntyre, C., Holmes, R. M., Zimov, N.
and Stubbins, A.: Detecting the signature of permafrost thaw in Arctic rivers, Geophys. Res.
Lett., 42(8), 2830–2835, 2015.

Sterner, R. and Elser, J.: Ecological Stoichiometry: The Biology of Elements from Molecules to
the Biosphere, Princetion University Press., 2002.

Sumner, L. W., Amberg, A., Barrett, D., Beale, M. H., Beger, R., Daykin, C. A., Fan, T. W.-M.,
Fiehn, O., Goodacre, R., Griffin, J. L., Hankemeier, T., Hardy, N., Harnly, J., Higashi, R., Kopka, J.,
Lane, A. N., Lindon, J. C., Marriott, P., Nicholls, A. W., Reily, M. D., Thaden, J. J. and Viant, M. R.:
Proposed minimum reporting standards for chemical analysis, Metabolomics, 3(3), 211–221,
2007.

t'Kindt, R., De Veylder, L., Storme, M., Deforce, D. and Van Bocxlaer, J.: LC-MS metabolic
profiling of Arabidopsis thaliana plant leaves and cell cultures: optimization of pre-LC-MS
procedure parameters., J. Chromatogr. B. Analyt. Technol. Biomed. Life Sci., 871(1), 37–43,
2008.

Talapatra, S. K. and Talapatra, B.: Shikimic Acid Pathway, in Chemistry of Plant Natural
Products, pp. 625–678, Springer Berlin Heidelberg, Berlin, Heidelberg., 2015.

Tfaily, M. M., Chu, R. K., Tolić, N., Roscioli, K. M., Anderton, C. R., Paša-Tolić, L., Robinson, E. W.
and Hess, N. J.: Advanced Solvent Based Methods for Molecular Characterization of Soil
Organic Matter by High-Resolution Mass Spectrometry, Anal. Chem., 87(10), 5206–5215, 2015.

Tholl, D., Boland, W., Hansel, A., Loreto, F., Röse, U. S. R. and Schnitzler, J.-P.: Practical
approaches to plant volatile analysis, Plant J., 45(4), 540–560, 2006.

Uzu, G., Sobanska, S., Sarret, G., Muñoz, M. and Dumat, C.: Foliar Lead Uptake by Lettuce

Exposed to Atmospheric Fallouts, Environ. Sci. Technol., 44(3), 1036–1042, 2010.

Vorholt, J. A.: Microbial life in the phyllosphere, Nat. Rev. Microbiol., 10(12), 828–840, 2012.

De Vos, R. C., Moco, S., Lommen, A., Keurentjes, J. J., Bino, R. J. and Hall, R. D.: Untargeted large-scale plant metabolomics using liquid chromatography coupled to mass spectrometry,

Nat. Protoc., 2(4), 778–791, 2007.

Walsh, M. C., Brennan, L., Malthouse, J. P. G., Roche, H. M. and Gibney, M. J.: Effect of acute dietary standardization on the urinary, plasma, and salivary metabolomic profiles of healthy humans., Am. J. Clin. Nutr., 84(3), 531–9, 2006.

Walther, G. R., Post, E., Convey, P., Menzel, A., Parmesan, C., Beebee, T. J. C., Fromentin, J. M.,

Hoegh-Guldberg, O. and Bairlein, F.: Ecological responses to recent climate change, Nature,

416(6879), 389–395, 2002.

Wang, R., Balkanski, Y., Bopp, L., Aumont, O., Boucher, O., Ciais, P., Gehlen, M., Peñuelas, J.,

Ethé, C., Hauglustaine, D., Li, B., Liu, J., Zhou, F. and Tao, S.: Influence of anthropogenic aerosol deposition on the relationship between oceanic productivity and warming, Geophys. Res. Lett.,

42(24), 10745–10754, 2015.

Warnes, G. R., Bolker, B., Bonebakker, L., Gentleman, R., Huber Andy Liaw, W., Lumley, T.,

Maechler, M., Magnusson, A., Moeller, S., Schwartz, M. and Venables, B.: gplots: various R

programing tools for plotting data, R package version 3.0.1., 2016.

Wedding, J. B., Carlson, R. W., Stukel, J. J. and Bazzaz, F. A.: Aerosol deposition on plant leaves,

Environ. Sci. Technol., 9(2), 151–153, 1975.

Whipps, J. M., Hand, P., Pink, D. and Bending, G. D.: Phyllosphere microbiology with special reference to diversity and plant genotype, J. Appl. Microbiol., 105(6), 1744–1755, 2008.

Wishart, D. S.: Metabolomics: applications to food science and nutrition research, Trends Food

Sci. Technol., 19(9), 482–493, 2008.

Wozniak, A. S., Bauer, J. E., Sleighter, R. L., Dickhut, R. M. and Hatcher, P. G.: Technical Note:

Molecular characterization of aerosol-derived water soluble organic carbon using ultrahigh resolution electrospray ionization Fourier transform ion cyclotron resonance mass spectrometry, Atmos. Chem. Phys., 8(17), 5099–5111, 2008.

Xiong, T.-T., Leveque, T., Austruy, A., Goix, S., Schreck, E., Dappe, V., Sobanska, S., Foucault, Y.

and Dumat, C.: Foliar uptake and metal(loid) bioaccessibility in vegetables exposed to particulate matter, Environ. Geochem. Health, 36(5), 897–909, 2014.

Zhang, A., Sun, H., Wang, P., Han, Y. and Wang, X.: Modern analytical techniques in metabolomics analysis, Analyst, 137(2), 293-300, 2012.

Zhang, Q., Stanier, C. O., Canagaratna, M. R., Jayne, J. T., Worsnop, D. R., Pandis, S. N. and

Jimenez, J. L.: Insights into the Chemistry of New Particle Formation and Growth Events in

Pittsburgh Based on Aerosol Mass Spectrometry, Environ. Sci. Technol., 38(18), 4797–4809,

2004.

**Table 1.** PERMANOVAs of the atmo-metabolome fingerprints generated by LC-MS, GC-MS and

FT-ICR instruments for overall metabolome comparison between seasons.

| LC-MS | | Sum of Squares | Mean Square | F | P |
|---|---|---|---|---|---|
| Season | 1 | 0.65 | 0.65 | 4.41 | 0.0001 |
| Residuals | 26 | 3.82 | 0.15 | | |
| Total | 27 | 4.47 | | | |
| GC-MS | | Sum of Squares | Mean Square | F | P |
| Season | 1 | 0.18 | 0.18 | 6.46 | 0.0003 |
| Residuals | 28 | 0.77 | 0.03 | | |
| Total | 29 | 0.94 | | | |
| FT-ICR | Df | Sum of Squares | Mean Square | F | P |
| Season | 1 | 0.1145 | 0.11 | 2.96 | 0.0285 |
| Residuals | 26 | 1.01 | 0.04 | | |
| Total | 27 | 1.12 | | | |

**Figure 1.** Schematic diagram showing the emissions of aerosols and posterior deposition on ecosystems.

[Figure]

**Figure 2.** Diagram of the main procedures of a general ecometabolomic study combined with complementary measurements.

[revised manuscript text omitted]

---

## Editor Comment (EC1) · P. Herckes (Editor) · 14 Feb 2017

The present manuscript describes an approach to characterize aerosols, using a suite of analytical techniques, with a goal of ecological studies. While the approach is novel for ecosystem studies, the methods are not novel in terms of atmospheric measurement techniques. No compelling case was made to address reviewer comments through the review iterations. Therefore, the manuscript does not fit the scope of Atmospheric Measurement Techniques and the decision is made at this stage to reject the manuscript for AMT.
* * *